# Human Circular Tourism as the Tourism of Tomorrow: The Role of Travellers in Achieving a More Sustainable and Circular Tourism

Martina Bosone [1,*] and Francesca Nocca [2,*]

1    Research Institute on Innovation and Services for Development of the National Research
     Council (CNR-IRISS), 80134 Naples, Italy
2    Department of Architecture, University of Naples Federico II, 80134 Naples, Italy
*    Correspondence: m.bosone@iriss.cnr.it (M.B.); francesca.nocca@unina.it (F.N.)

**Abstract:** Nowadays, the challenge of sustainability is increasingly played out in cities, which represent the favoured field of action to implement strategies and actions for supporting the transition towards a more human and ecological development paradigm. The problems caused by the current economic model (linear model) are even more stressed today due to the effects produced by the COVID-19 pandemic. The tourism sector (one of the world's major economic sectors and, thus, one of the main players in the development dynamics) is one of the economic sectors that has been the most negatively impacted by the pandemic. In this study, Human Circular Tourism (HCT) is proposed as a strategy to move towards a more sustainable future and, thus, reduce the negative impacts produced by the tourism sector. In particular, the objective of this paper is to understand the awareness of travellers (one of the categories of actors involved in the tourism experience) regarding sustainable and circular tourism in order to support local governments in the elaboration and implementation of strategies and actions towards more sustainable and circular tourism. To this end, a survey was conducted. In particular, a questionnaire was developed and submitted to a sample of tourists from all over the world to understand their behaviours and perceptions in their tourist experiences. From a critical analysis of the results, it emerges that there is a growing awareness of issues related to the concept of sustainability, especially in relation to the major issues of climate change and people's health. This perception has certainly been influenced by the health emergency from COVID-19, but the sample of interviewees reveals that much still needs to be invested in increasing their awareness of the complexity of the factors involved in more sustainable, circular, and human-centred tourism. Therefore, starting from this, possible future prospects for the tourism sector from the circular economy perspective are here identified.

**Keywords:** sustainable tourism; Human Circular Tourism; COVID-19

## 1. Introduction

Nowadays, the challenge of sustainability is increasingly played out in cities, which represent the favoured field of action to implement strategies and actions for supporting the transition towards a more human and ecological development paradigm.

Problems such as air and water pollution, land consumption, and climate-changing gas emissions, which are strongly present in our cities, are already showing their negative effects on human health and natural ecosystems, with implications also in economic sectors [1].

The problems caused by the current economic model (linear model) are even more stressed today due to the effects produced by the COVID-19 pandemic. In fact, the latter, in addition to having caused serious damage to health, has also radically changed habits, behaviours, and social relationships and caused negative impacts that affect not only the health sector but the various sectors of the economy.

However, on the other side of the coin, the pandemic due to COVID-19 has produced positive effects on the environment linked to reduced travel, the interruption of certain activities, and limitations imposed by the government.

Among the different sectors of the economy, one that has most suffered the negative impacts of the pandemic is tourism [2,3], which is one of the world's major economic sectors (the third largest socio-economic activity in Europe [4] and therefore has a considerable weight in the development dynamics. Tourism is the third-largest export category (after fuels and chemicals). For some countries, it can represent over 20% of their GDP and, overall, it is the third largest export sector of the global economy.

In 2019, the tourism sector accounted for 7% of global trade, representing the third largest export sector of the global economy and over 30% of exports for the majority of SIDS (Small Island Developing States), 80% in some cases [5].

The data published by UNWTO (UNWTO, 2019) highlight that the COVID-19 pandemic has influenced all areas of the tourism value chain, which, in a pre-COVID situation, produced many positive impacts at different levels. Indeed, the UNWTO also highlighted the interdependence between tourism and aspects related to preserving the planet and mitigating impacts on natural and cultural ecosystems. In fact, on an environmental level, tourism plays a fundamental role in influencing the ecosystem balance, not only because of the impacts it produces in terms of emissions and pollution but also because of its ability to generate economies based on the exploitation of natural resources. This happens especially in SIDS (Small Island Developing States) and LDCs (Least Developed Countries) (e.g., in Africa), in which environmental resources are the main tourist attraction (up to 80% of visits), and the resulting revenues are reinvested in biodiversity conservation. In general, wildlife tourism represents 7% of world tourism and is a growing sector (3% per year).

Tourism is also an important sector for the employment opportunities it offers: It supports 1 in 10 jobs [6,7] and provides livelihoods for many more millions of people.

Tourism (and related activities) contributed to 10.4% of global GDP in 2019 [8]; international tourist arrivals grew by 5% in 2018 to reach the 1.4 billion mark and, at the same time, export earnings generated by tourism have grown to USD 1.7 trillion. As the above data show, this sector generates positive impacts in terms of employment, export revenues, etc. This underlines the importance that this sector can play in post-COVID-19 economic recovery.

However, in addition to the aforementioned positive impacts, this sector generally contributes to a number of negative environmental and social impacts [9,10] because it is configured according to the model of linear economy [11]. It can put enormous pressure on an area and produce impacts such as soil erosion, increased pollution, discharges into the sea, natural habitat loss, increased pressure on endangered species, and heightened vulnerability to forest fires [12]. According to Lenzen et al. (the latest estimation about global carbon emissions related to tourism) [13], between 2009 and 2013, it contributed to 4.6% of global warming, with an increase in global carbon footprint from 3.9 to 4.5 GtCO2e, equal to 8% of global greenhouse gas emissions. Furthermore, all activities linked to tourism, such as transport, shopping, and food, are significant contributors to environmental pollution [13]. In 2016, $CO_2$ emissions from the transport-related tourism sector accounted for about 22% of total transport emissions and 5% of overall man-made emissions (1.3 points more considering the value of 3.7% in 2005), out of which 64% were caused by passenger transport. In 2030, the total transport-related tourism emissions (excluding cruises) are expected to grow to make up to 23% of transport emissions, equal to 5.3% of the overall forecast of man-made emissions [14]. The analysis conducted by UNWTO from 2005 [15] to today highlighted that from 2005 to 2016 there was an increase of 62% in transport-related $CO_2$ attributable to tourism, which is expected to experience another increase of 25% compared with 2016 in the forecasts for 2030 [14]. This aspect adds more challenges to the tourism sector's ambition to meet the targets of the Paris Agreement [16].

In particular, the air travel sector represents one of the main tourism contributors to global warming [14], as it was responsible for 2.9% of global $CO_2$ emissions in 2020 [17] and

for 3.5% of effective radiative forcing in 2018 [18]. The forecasts for 2030 highlight the role of this sector, which is expected to remain absolutely relevant in international tourism [14].

Services used in the hospitality sector (e.g., air conditioning, heating, restaurants, etc.) account for about 20% of emissions [15,19].

As outlined above, the tourism sector plays a key role in development processes on a global scale. However, the aforementioned positive impacts (i.e., employment, biodiversity conservation, local economy valorisation) have been reversed in the last two years due to the pandemic (see Section 2).

The crisis due to COVID-19 has also demonstrated that dividing the three dimensions (ecological, economic, and social dimensions) has been a huge mistake. It forces us today to consider a logic for the current economy linked to that of ecology and society.

Considering that the number of tourists travelling across borders is expected to reach 1.8 billion a year by 2030 [20], this will bring many opportunities on the social and economic levels, but, at the same time, it will contribute to the increase in environmentally negative impacts. Due to the important role of the tourism sector on the socio-cultural, economic, and environmental levels, it can play an important role in sustainable urban transformation.

The objective of this paper is to support and orient the identification strategies for achieving more sustainable tourism. In particular, in this paper, Human Circular Tourism (HCT) (so called by the authors) is proposed as a strategy able to support a transition towards a more sustainable future in the tourism sector.

Since the success (or failure) of strategy implementation depends on the different stakeholders, in this study, a survey was conducted. In particular, a questionnaire was developed and submitted to a sample of tourists from all over the world to understand their behaviours and perceptions in their tourist experiences to better orient possible future prospects for the tourism sector from the circular economy perspective.

The paper is organised as follows: after an introduction about tourism and the impacts of COVID-19 on this sector (Section 2), its contribution to the Sustainable Development Goals is analysed (Section 3). In Section 4, the methodology is explained, and Human Circular Tourism (HCT) is proposed as a strategy to operationalise the principles of Sustainable Tourism. In particular, the development of a survey to understand travellers' behaviours is conducted. The results of the survey are analysed in Section 4.2 and discussed in Section 5 to identify possible future perspectives of the tourism sector. Finally, Section 6 presents the strengths and limitations of the proposed approach and future research perspectives.

## 2. Theoretical Background: Tourism and COVID-19

As stated in the previous paragraph, the crisis due to COVID-19 caused profound changes in socio-economic dynamics. Tourism is one of the sectors that suffered a sudden, global, and abrupt shock to the demand.

The need to reorganise the whole tourism sector can be interpreted as an opportunity to reorient international strategies towards a carbon neutral and resilient tourism economy, one that is also able to consider its interactions with and effects on societies, other economic sectors, and cultural and natural resources.

The data published by UNWTO [21,22] highlight that the COVID-19 pandemic has influenced all areas of the tourism value chain, which is able to produce both positive and negative impacts on different levels due to its interdependence with the preservation of the planet and the mitigation of impacts on natural and cultural ecosystems.

On the environmental level, tourism plays a fundamental role in influencing the ecosystem balance, not only in terms of emissions and pollution but also in terms of exploitation of natural resources, which are often a source of tourist attraction and on which many economies were based.

From a socio-cultural point of view, many experiences of community-based tourism show that the involvement of communities in the protection and enhancement of their cultural and natural heritage contributes to the enhancement of local living conditions in terms of wealth and well-being.

All these aspects were affected in all countries and on all levels due to the COVID-19 crisis. Since the beginning of the pandemic, the UNWTO has been developing studies and analyses to assess the main impacts of COVID-19 on tourism and to prefigure recovery scenarios [23]. According to the aforementioned studies, although global tourism picked up 4% in 2021 compared with 2020 (415 million in 2021 vs. 400 million in 2020), the negative impacts of COVID-19 led to a 72% decrease in tourist arrivals compared to 2019. The absolute worst year for tourism remains 2020, when international arrivals fell by 73% [24].

The drop in international tourist arrivals resulted in an estimated loss of USD 1 trillion in export revenues with a negative impact on the global GDP of USD 1.6 trillion in 2021 [24]. However, in 2021, the direct gross domestic product of tourism was USD 1.9 trillion, higher than the USD 1.6 trillion in 2020, but still well below the pre-pandemic figure of USD 3.5 trillion (in 2019).

According to the UNWTO report on travel restrictions related to COVID-19, the more or less restrictive measures adopted in the different tourist destinations are also manifested in the choice to require a negative molecular test or antigen test upon arrival [23].

The spread of the vaccine, the elimination of numerous travel restrictions, greater coordination, and clearer information on travel protocols are the main factors identified by experts for an effective recovery of international tourism, even though the recent increase in cases of COVID-19 and the new variants are factors that continue to disrupt recovery and influence both travellers' and governments' choices on whether or not to strengthen travel bans and restrictions [24].

Considering a tourism recovery strategy on a global scale, one of the main issues is related to how to protect the 100 million jobs at risk (directly linked to the tourism sector) and also the jobs in sectors associated with tourism (i.e., accommodation, food, and others, which employ 144 million people worldwide).

The same problem arises with regard to the development and adoption of support measures for small businesses (which account for 80% of global tourism), which have proved particularly vulnerable in this particular period.

The pandemic has not only influenced the tourist employment sector, but has also had important economic, socio-cultural, and environmental consequences for tourism-dependent communities. These include the closure of local handicraft markets, the interruption of oral traditions and festivals (intangible cultural heritage), a decrease in economic support for biodiversity conservation (in the case of natural heritage), and an increase in "poaching" phenomena as a result of reduced tourist flows and staff presence [25,26].

Based on these considerations, it is clear that the COVID-19 crisis represents a watershed moment to be interpreted as an opportunity to reorient collective action and international cooperation towards more sustainable, inclusive, and carbon-neutral tourism.

The UNWTO Policy Brief [27] identifies five priority areas for recovering the tourism sector: to mitigate socio-economic impacts on livelihoods (ensuring gender equality in employment), boost competitiveness and build resilience (in particular, reduce the vulnerability of micro-, small-, and medium-sized enterprises), advance innovation and the digital transformation of tourism (creating new job opportunities based on digital skills), foster sustainability and green growth, and stimulate coordination and partnerships. Thus, the UNWTO Policy Brief stresses the ability of tourism to produce economic, social, cultural, and environmental impacts.

In view of the above, new development models are needed to reduce the negative impacts of tourism and, at the same time, amplify the benefits it can produce in post-COVID-19 recovery.

## 3. Literature Review

*The Contribution of the Tourism Sector to the Sustainable Development Goals (SDGs)*

Many studies highlight the relationship between the tourism sector and the other sectors that are directly or indirectly linked to it, on which it produces impacts.

However, while the contribution of the tourism sector to the SDGs is clear from a theoretical point of view [28], there are few studies that analyse it from a more practical point of view [29], as the operational aspect of this link requires a longer time for it to be meaningfully reflected in practice, allowing for the results to be read.

As highlighted by many studies [11,30,31], the tourism sector plays a crucial role in achieving many Sustainable Development Goals (SDGs) [32]. The SDGs are key elements in the United Nations' Agenda 2030 and represent the effort to steer the definition of a strategy "to achieve a better and more sustainable future for all" [33], starting from the recognition of the close link between human well-being, the health of natural systems, and the presence of common challenges for all countries, such as peace, development, and human rights. In addition, the definition of specific goals and targets was intended to facilitate the monitoring of progress implemented by states in order to reason based on measurable data, objectives, and targets [34].

The SDGs in which the tourism sector is explicitly mentioned are the following three [30,31]:

- Goal 8, "Decent work and economic growth" (in particular, target 8.9). Tourism aims to create jobs and promote local culture and products through the elaboration and implementation of policies promoting sustainable tourism.
- Goal 12, "Responsible consumption and production" (in particular, target 12.b). This goal refers to identifying development models and strategies for implementing sustainable tourism. In particular, it refers to the development and implementation of tools to monitor sustainable development impacts from sustainable tourism that create jobs and promote local culture and products.
- Goal 14, "Life below water" (in particular, target 14.7). This is related to the conservation and preservation of fragile marine ecosystems, in particular for Small Island Developing States (SIDS) and LDCs (Least Development Countries), as coastal and maritime tourism are the biggest segments in this sector. The sustainable management of fisheries, aquaculture, and tourism will be a useful way to promote a blue economy.

In addition to the above three SDGs, which are directly linked to the tourism issue, the other SDGs can also be connected to tourism. In fact, as UNWTO has highlighted [30], tourism can contribute to improving the quality of life and well-being of communities, creating new skills and new employment opportunities (Goal 3, "Good health and well-being"; Goal 4, "Quality education"; and Goal 5, "Gender equality") [35], stimulating the sale and consumption of local products in the agricultural sector (Goal 2 "Zero hunger"), and improving communities' incomes through the development of small business and entrepreneurship (Goal 1, "No poverty"; Goal 3, "Good health and well-being").

In particular, for Goal 1, "No poverty", many international institutions (UNWTO, the World Travel and Tourism Council—WTTC, the World Bank, the World Trade Organization, and the International Monetary Fund—IMF) have launched several development programs (i.e., the Sustainable Tourism for Poverty Eradication Program), identifying tourism as a tool for economic development for less developed countries (LDCs) and as an opportunity in the international market [36]. The reasons why tourism can bring benefits to less developed countries are of different types and interrelated [37], although they are still supported by only a few empirical studies [37,38]: The wealth of cultural and natural heritage present in less developed areas, where most of the poor population is concentrated, represent great potential in terms of tourism supply and, at the same time, would require a large amount of skilled labour. This aspect for local communities would constitute both a job opportunity and the development of new skills, favouring the creation of small and micro-entrepreneurs. Moreover, the fact that the place of consumption coincides with the place of production represents an opportunity for direct interaction between tourists and local stakeholders, contributing to the support of the local economy and, consequently, to the improvement of existing infrastructures and the preservation of natural and cultural heritage. All these aspects, based not only on the improvement of the living conditions of local communities but also on their empowerment, would also positively influence their sense of belonging

to the place and community and increase their awareness of the value of their cultural heritage, contributing to making cities and human settlements more inclusive, safe, resilient, and sustainable (Goal 11, "Sustainable cities and communities").

For Goal 3, "Good health and well-being", energy, health, and wealth are the key factors of sustainable tourism, which are capable of ensuring the improvement of well-being conditions and, thus, the quality of life, as well as, at the same time, the preservation of the planet [39].

Tourism can influence health in different ways and, above all, affects both consumers and producers. In fact, health and safety are fundamental preconditions not only in influencing tourists' choices about destinations and travel conditions (conditioned, for example, by the epidemiology or availability and quality of health services in the host country, hygienic conditions, water quality, etc.) but also affecting the health conditions of the host population [40].

In fact, tourist flows can upset certain balances in the local population from different points of view.

First of all, some studies [41] argue that, especially in the case of tourist destinations located in underdeveloped countries, the financial disparity between travellers and guests can lead to strong cultural changes. In fact, host communities, perceiving such "distance", interpret tourists as the embodiment of progress, wealth, and a desirable lifestyle and, consequently, are led to transform their habits to get closer to that model. This mechanism, besides representing cultural erosion, also favours greater promiscuity, increasing the risk of infection by sexually transmitted diseases.

Secondly, the increase in international travel makes the risk of the rapid spread of infections more likely, especially in the event of sudden emergencies. In addition to this factor, the increasing possibility of being able to reach previously unreachable destinations increases the risk of contracting infections caused by poor hygienic conditions in these places. While this trend could stimulate the development of treatments and vaccinations in parts of the world that were previously of little economic interest to pharmaceutical companies, these treatments should be targeted not only at tourists but, above all, at the local populations at risk. Ensuring universal and equitable access to healthcare requires efforts to reduce inequalities (Goal 10, "Reduced inequalities") and discrimination of all kinds (Goal 5 "Gender equality"). One solution, for example, could be for governments to invest the revenue generated by tourism in good quality healthcare systems that are accessible and affordable for all in order to avoid further privatisation and the inflation of medical costs. Alternatively, governments could establish compensatory forms whereby, for every foreigner treated in the tourist destination, a percentage of local people are guaranteed access to the local healthcare system.

Another aspect that concerns the safety and health of host populations is linked to the job opportunities offered by tourism. In fact, if they do not comply with health and safety standards, they have harmful effects on the physical and mental health of workers (Goal 8 "Decent work and economic growth"). Tourists should be the first to critically evaluate their consumption behaviour and make sure that it does not infringe on the right of the local population to a healthy and dignified life [42–44]. In addition, there is a growing demand for specific health-and-wellness-related tourist destinations. The rediscovery of these ancient traditions can represent a great opportunity to activate small-scale authentic tourism circuits based on local know-how and the involvement of local communities in recognised and justly remunerated forms of employment.

The promotion of the health and well-being of all is also linked to other aspects, such as the achievement of food security (Goal 2 "Zero hunger") [45–47], clean water and sanitation (Goal 6) [48–50], or functioning ecosystems (Goal 14 "Life below water", Goal 15 "Life on land") [51–53]. In fact, tourism plays a key role in improving the environment through, for example, the adoption of sustainable energy systems (Goal 7 "Affordable and clean energy") [39], helping to mitigate climate change by lowering energy consumption and

shifting to renewable energy sources (Goal 13 "Climate action") [34,54,55], and preserving biodiversity (Goal 15 "life on land") [34,35].

With particular reference to the achievement of Goal 13 of the SDGs, which is about climate change, the relationship between the tourism sector and climate change should be highlighted, in which the tourism sector is, at the same time, both "a vector and a victim" of climate change [56]. Research and studies about the relationship between climate change and tourism began in the 1960s [57], and in recent times, many international organisations [30,31,58–60] have tried to systematise the frameworks, tools, and practices on tourism and climate change adaptation and mitigation.

Tourism can produce negative impacts: It can cause environmental damage and pollution, the degradation of heritage, various waste (e.g., in the renovation industry), etc., because it is configured according to a linear economy model (a "take–make–dispose" model). Certainly, the tourism sector is capable of producing wealth in the short term (from trade to employment, etc.), but the net benefits may be smaller when considering direct and indirect costs (i.e., environmental impacts).

As already pointed out, tourism is a sector that, more than the others, is characterised by the linear economy model, and thus, it is one of the most responsible for climate change. In fact, after the intensive tourist "utilisation" of a site, once a certain threshold of exploitation has passed, only a "set of waste materials" [61] remains.

It is, therefore, necessary to put the process of tourism development into a different perspective. It is necessary to identify new tourism management and development strategies that can produce greater benefits and, at the same time, reduce costs (environmental, social, and economic costs). From this perspective, the circular economy is proposed as a model that can help make tourism more sustainable (see Section 4.1).

Therefore, climate change and tourism are two mutually influencing factors: On the one hand, climate change affects the usability of tourist destinations (in terms of attractiveness and the quality of the environment) and determines the trend of tourist flows; on the other hand, tourism is one of the main factors affecting the climate (i.e., due to the increase in emissions from transport, increased waste production and energy consumption related to accommodation and tourist flows, etc.). Climate change has negative effects, especially on tourist destinations whose main attractions are represented by particularly vulnerable natural resources (such as small islands or coastal areas) and where tourism represents the main economic resource for local communities. In fact, in these contexts, the occurrence of catastrophic events, in addition to compromising the balance of the natural ecosystem, also has obvious impacts on tourism and related activities, slowing down and sometimes blocking the recovery of the affected places and economies linked to tourism.

The health emergency caused by COVID-19 has further highlighted the link between pollution, health, and climate change. From May to September 2020, the World Health Organization (WHO) stressed the above issue, particularly in the "Manifesto for a healthy recovery from COVID-19", launched in May 2020. This document highlights the importance of protecting and preserving nature as a source of human health (first prescription). Six WHO prescriptions and a comprehensive set of key actions for achieving healthier environments are provided [62]. In this framework, the contribution of the tourism sector can be significant.

As mentioned above, tourism is cross-sectorial and, for this reason, its sustainable implementation requires public/public cooperation and public/private partnerships through the engagement of multiple stakeholders (Goal 17, "Partnerships for the goals"), also favouring the knowledge and the integration among people of diverse cultural backgrounds (Goal 16, "Peace, justice and strong institutions").

However, the impacts of COVID-19 threaten to roll back progress made in advancing the Sustainable Development Goals (SDGs), making it necessary to identify models and tools able to support the transition towards a healthier, fairer, and greener world.

## 4. Materials and Methods

In this study, the circular tourism model is proposed as a model for making the tourism sector more sustainable and capable of producing environmental, social, and economic benefits at the same time (Figure 1). Thus, in the following paragraphs, this model is explained and critically analysed.

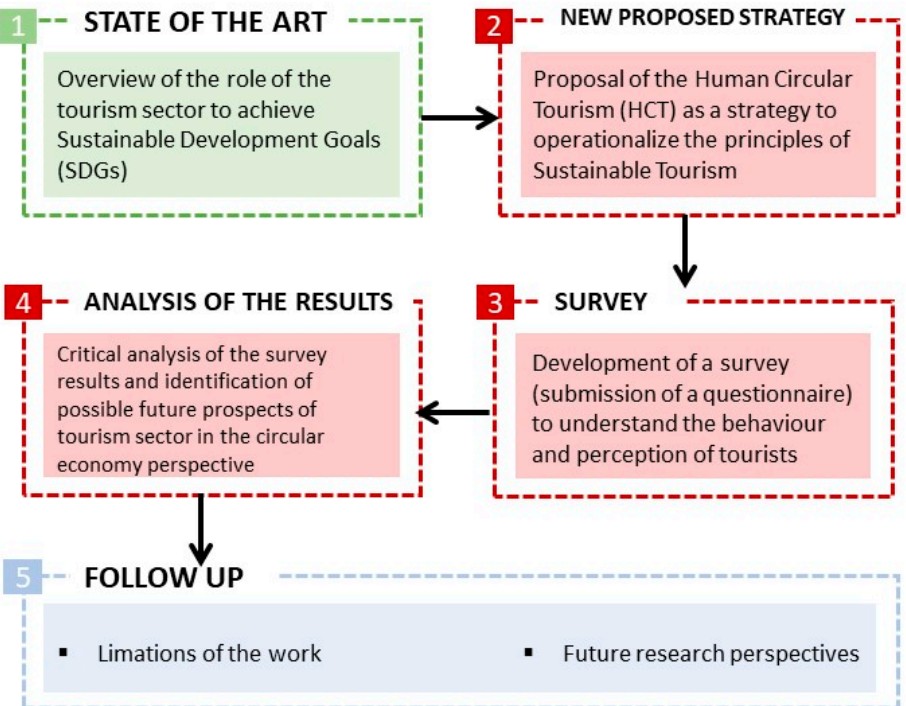

**Figure 1.** The methodological workflow.

However, the circular tourism model, in order to be successful, requires a strong cultural base from all actors and stakeholders involved in the tourism experience. This cultural base, which determines people's behaviour, can be oriented and strengthened through appropriate policies and strategies implemented by governments.

Starting from the awareness that the success (or not) of the transition towards circular and more sustainable tourism greatly depends on the mindset of travellers, this study aims to understand their point of view on sustainable and circular tourism, their awareness, and their actual behaviour in this perspective in order to identify and orient policies and actions in this sector to move towards more sustainable tourism. The aim is to understand if tourists today assume "circular behaviour" and, if not, to understand what aspects need to be strengthened and on which to base strategies and actions to move towards circular tourism.

In this study, a survey was conducted involving communities through interviews aimed at understanding their awareness of sustainable and circular tourism, their behaviours as tourists, and the influence of COVID-19 on travellers' choices (Figure 1).

To this end, a questionnaire was developed and submitted via the Google Forms online tool. The survey included different phases: the definition of the objective of the survey, the identification of stakeholders, the development of the questionnaire, the submission of the survey, and the deduction and critical analysis of the results.

The questionnaire was distributed to the community directly (contacting by email) or through social networks. A total of 216 questionnaires were filled in in a time span of three months (2021).

The questionnaire is divided into four sections. All the questions are structured as multiple-choice answers.

After an introduction about the object and aim of the questionnaire, the first section is related to the collection of the respondents' data in order to understand the sample analysed.

The second section, "To be a sustainable tourist", aims to understand the level of awareness of interviewees about the concept of sustainable tourism and what their behaviours are as travellers, that is, if they adopt "circular behaviours" or not.

The third section, "To be an active and responsible tourist", is focused on the economic and socio-cultural aspects of the tourism sector in order to understand the relationships that can be established between travellers, places, and host communities on different levels.

The last section of the questionnaire, "Tourism and COVID-19", is focused on understanding the influence of COVID-19 on tourists' behaviours and their awareness of the relationship between the pandemic and environmental issues.

### 4.1. Human Circular Tourism (HCT) for Achieving a More Sustainable Future

In order to make the tourism sector more sustainable, new models and tools are fundamental. In this study, Human Circular Tourism (HTC) is proposed to achieve this goal. HCT can also support the post-COVID-19 recovery of that sector.

The circular tourism model, today, is scarcely analysed in the scientific literature [63] and is often exclusively related to environmental and eco-friendly issues [64–68]. Indeed, it is a wider concept.

Here, circular tourism is interpreted as "the tourism that transforms its processes from linear (take-make-dispose) to circular (take-make-use-remake) ones" [11]. It limits impacts on the environment, and in which actors of tourism (traveller, host, tour operator, supplier) adopt an eco-friendly and responsible approach [58]. The circular tourism sector is referred to as such due to its capacity to trigger and stimulate circular flows, aiming to conciliate the tourism sector and sustainable resource management. However, that is not all. Circular tourism is not just green tourism, addressed to limit the consumption and waste of non-renewable energy resources. Recovery, reuse, and redevelopment, but also valorisation and regeneration, are key words if we think about when considering sustainable and circular tourism. We can "use" tourism as a means to regenerate knowledge produced by each territory in terms of values, language, significance, and skills. Functional reuse not only refers to fixed capital but also to knowledge and values. From this perspective, circular tourism represents a means to fix memory in the era of the "instant", of the "hic et nunc". Through functional reuse, we are able to regenerate values, keeping them "in time" [11].

There are some international good practices from the perspective of circular tourism. They are mainly related to the hospitality and mobility sectors.

Dutch Hospitality in the Netherlands [69], which turns coffee grounds from its restaurant into oyster mushrooms and redistributes them to its restaurant, is an example of this direction. It turns a waste product into a nutrient, perfectly in line with the principle of closing the cycles of the circular economy.

The Pakasai Resort (Krabi Province, Thailand) was awarded the ASEAN Green Hotel Award 2014 due to its ecological commitment to all activities and services offered by the hotel (https://www.pakasai.com/sustainability/; accessed on 9 July 2022). The rooms are sanitised only with environmentally friendly products, they are equipped with energy-saving appliances and LED lighting, and the supply of personal hygiene products is totally plastic-free. Recovered grey water is reused for the flushing system in the rooms and to irrigate green spaces. The facility staff meets weekly to assess whether the performance of their services is contributing to meeting sustainability targets. More generally, this resort aims to minimise its impact on the surrounding ecosystems and to produce benefits for the local community through collaboration with local organisations. For example, the Resort Garbage Bank financed the construction of a water tank for the local village.

The Siloso Beach Resort (https://www.silosobeachresort.com/; accessed on 25 June 2022), on the southwest coast of Sentosa Island (Singapore's south coast), is an award-winning eco-resort that integrates the surrounding habitat into its design, favouring open spaces by protecting the original trees and planting new ones and using natural springs to feed

a landscaped swimming pool built following the morphology of the land. Moreover, the resort offers services with a reduced ecological impact (locally sourced food, limited use of plastic, and reduced energy consumption) and proposes eco-friendly activities (eco-adventures, cycle tours, and hikes).

Another example is the Ladybird Farm Leisure Hotel (in Hungary), which allows its guests to pay part of their entrance fee with recyclable household waste, according to the "waste = money" principle [70].

The widespread hotel concept (e.g., in Matera, Italy) is also in line with the principles of circular tourism, as it focuses on the recovery, conservation, and enhancement of the territory and its traditions and peculiarities [71].

The transition to this new model of tourism is only possible if all stakeholders and actors really understand the benefits and, consequently, change their (possible) bad behaviours. All subjects (accommodation owners, tourism industry staff, tourism service providers, the tourists themselves, etc.) are responsible for this transition, which necessarily requires a modification of lifestyles and behaviours. This change depends on the level of people's awareness about this issue [72,73]. A "cultural revolution" is, therefore, necessary [11].

The UNWTO identifies different categories of subjects involved in the tourism sector: traveller, public body, international organisation, donor, academia, and CSO company. They are put in relationships in the "Tourism4SDGs.org" platform [58], which highlights the relationship between the SDGs and the tourism sector, providing the global tourism community with a space to co-create and engage to implement the 2030 Agenda for Sustainable Development and identify recommendations and actions for each category.

Most studies of circular tourism focus on implementing the circular economy model on the supply side and not on the demand side [74–77]. They mainly pursue a reduction in waste and pollutants from the tourism industry.

Few studies, however, focus on the role of the tourist/traveller [78–82]. This is the reason why a survey was carried out in this paper, which was used to understand the level of awareness of travellers on this issue, considering that the success or failure of the implementation of this model also depends on their behaviour.

The ultimate goal of sustainable development and, thus, strategies to achieve it (such as HCT), is the human being. Therefore, such development models have to be aimed at guaranteeing and promoting the rights and needs of human beings from an intergenerational perspective.

Furthermore, the operationalisation of sustainable human development presupposes the direct and effective involvement of people, in the definition, implementation, and evaluation of the various choices. The human dimension of development not only separates but integrates and valorises each component from a systemic perspective, stimulating and strengthening not only human/nature but also human/human relations. The results of the survey (in the following section) can provide significant support for orienting government strategies and actions in this direction.

### 4.2. Results of the Survey

At the end of November 2021, the questionnaire was closed and the results were analysed. The questionnaire was sent to 250 people. In total, 216 completed questionnaires were returned. The results of the questionnaire were analysed by means of graphs and cross-tabulation analysis in order to understand the awareness and behaviours of travellers with respect to sustainable and circular tourism and in relation to COVID-19 impacts. Starting from the results, the other steps of this study were elaborated. In fact, these analyses were carried out to deduce suggestions to support and orient government strategies and actions.

In order to simplify the presentation of the results, this section has been divided into different sub-sections corresponding to the different sections of the questionnaire.

At the beginning of each sub-section, the results are summarised in a table and then analysed more in-depth. The reference to the "pre" and "post" COVID-19 phase is only a

part of the analysis, which aims to take a snapshot of the change that occurred abruptly. All the other reflections from the survey refer to aspects that occur independently of COVID-19.

The responding sample (first section of the questionnaire) is quite young (probably because the dissemination of the questionnaire via the Internet could reach young people more easily). In fact, 39.8% of the respondents are between 25–34 years old and 21.8% are between 35 and 44 years old. Most of them are employed or self-employed (Figure 2).

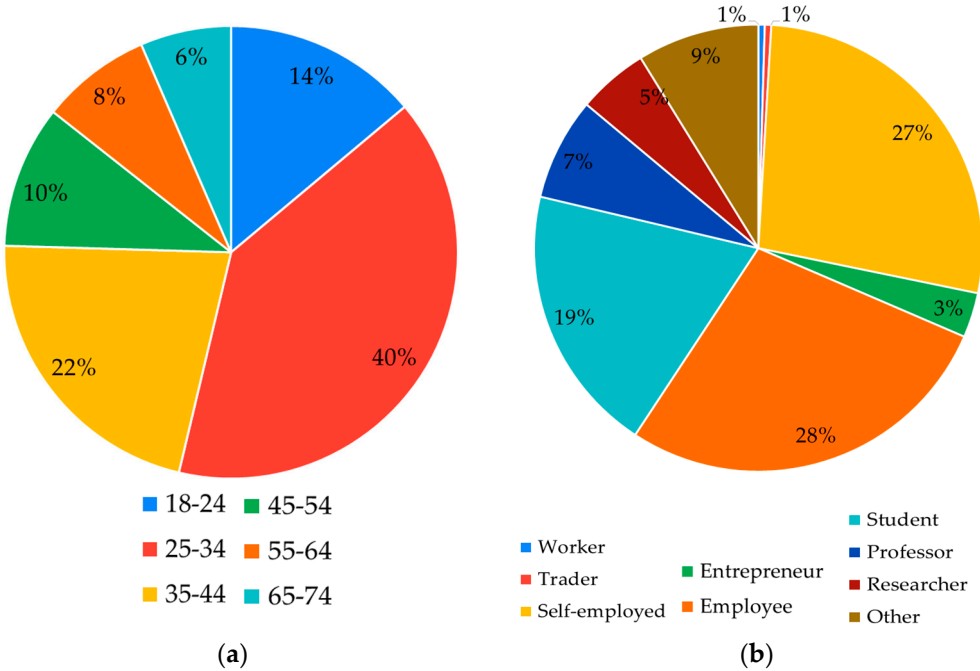

**Figure 2.** (**a**) Respondents' ages; (**b**) respondents' jobs.

### 4.2.1. Be a "Sustainable" Tourist

The "Be a 'sustainable' tourist" section is about interviewees' perceptions and behaviours in relation to sustainability issues in the tourism sector. The results are summarised in Table 1 and then analysed more in-depth with the support of many graphs. This section is more focused on environmental sustainability (except for the first two questions), while the following sections investigate the social and cultural aspects.

Figure 3a shows the respondents' travel motivations. Most of them (60.6%) stated that the main reasons for their trips are related to cultural experiences. However, 16.4% of them travel mainly for wellness-related reasons, 8.9% for work, and 6.6% to seek contact with nature, and nobody declared "sport" or "knowledge of new places" as travel motivations (Figure 2).

**Table 1.** Summary of the results of the second survey section: "Be a 'sustainable' tourist".

| Be a "Sustainable" Tourist | |
|---|---|
| **Issue** | **Result** |
| Respondents' travel motivation | Cultural experience is the main reason for their trips, followed by wellness-related reasons, |
| Factors expected by respondents from sustainable tourism | Enhancing the local heritage (cultural, natural, food, wine, etc.) and reducing environmental impact. |
| Most frequent behaviours adopted by tourists during the travel | - Buy typical products of the place visited;<br>- Pay attention to reducing energy and water consumption. |

**Table 1.** *Cont.*

| Be a "Sustainable" Tourist | |
|---|---|
| **Issue** | **Result** |
| Relationship between factors that respondents attribute to sustainable tourism and the behaviour they actually adopt | Respondents who linked sustainable tourism to a reduction in environmental impacts:<br><br>- Adopt behaviours that respect the local cultural heritage;<br>- Buy typical local products;<br>- Pay attention to reducing energy consumption;<br>- Separate waste;<br>- Carry a water bottle during the trip. |
| Attention paid to avoiding waste at the tourist facilities | Respondents who linked sustainable tourism to the enhancement of the local cultural heritage:<br><br>- Separate waste;<br>- Carry a water bottle during the trip. |
| | The majority of respondents are "moderately" or "extremely" attentive in avoiding producing waste. |
| Relationship between respondents' attention to avoiding waste in accommodations and their age | - The 25–34-year-olds had a high level of attention to avoiding waste at the facilities where they stay;<br>- In the 55–64 and 65–74 age groups, nobody declared themself to be "not at all" attentive to avoiding waste;<br>- In the 55–64 and 65–74 age groups, the majority of respondents stated they were "extremely" attentive to avoiding waste. |
| Influence of the presence of sustainable services in the choice of the tourist facility | About 75% of respondents split between those for whom it was "moderately" important and "not at all" important. |
| Means of transport used in travel destination | The majority of respondents prefer public transport.<br>The second most favoured mean of transport was walking. The minority of respondents consider the possibility of moving by car-sharing services. |
| Factors influencing the choice to purchase products during travel | The majority of respondents prefer goods identified with the destination's brand. |

Figure 3b shows the results of a multiple choice question about the factors expected by respondents from sustainable tourism. For the majority of respondents (162 out of 216), the concept of sustainable tourism is closely linked to activities and actions aimed at enhancing the local heritage (cultural, natural, food and wine, etc.), at reducing environmental impact (141 out of 216), at establishing a cultural exchange with the local population (69 out of 216), and, finally, activities and actions with low energy consumption (65 out of 216). A minor percentage of respondents (42 out of 216) link the concept of sustainable tourism to a relationship with nature.

Figure 4 shows different kinds of behaviour adopted by tourists during their travel. Since this was a multiple-choice question, the type of graph we used is useful in showing the frequency of the answers given by the respondents, also allowing us to make reflections on which aspects are most interrelated. The results show that for respondents it is important to adopt behaviours that respect the cultural and natural heritage of the place visited. The second most frequent behaviour among respondents is to buy typical products of the place visited. The importance attributed to aspects more closely linked to the purely environmental dimension also emerges: respondents declared they were careful about energy and water consumption during their tourist experiences, and half of them tend not to print booking receipts (travel, hotel, etc.) to avoid waste, saving them digitally on electronic devices. The least frequently adopted behaviour is to choose accommodation with environmental certifications.

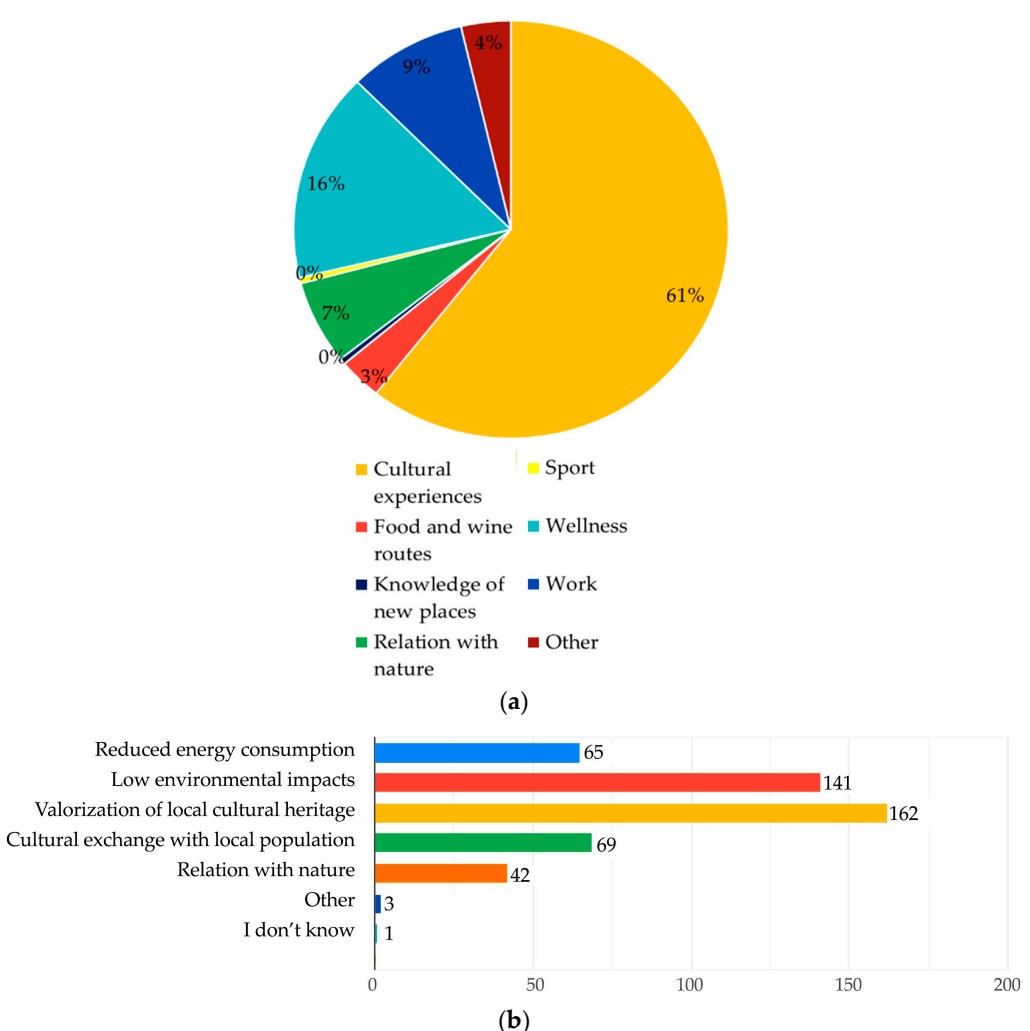

(**a**)

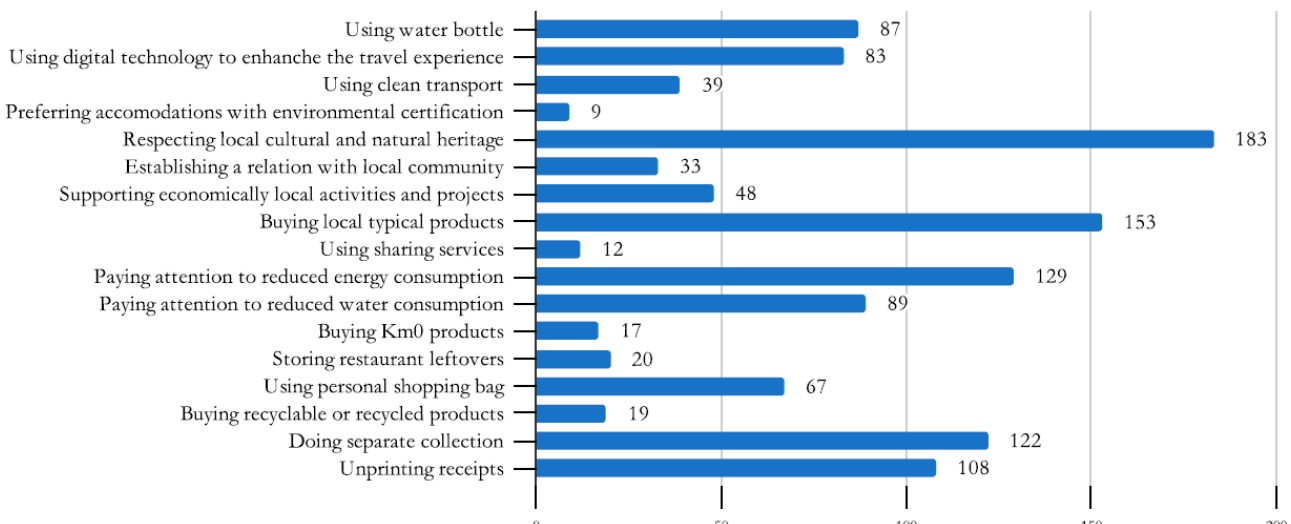

(**b**)

**Figure 3.** (**a**) Respondents' travel motivation; (**b**) factors expected by respondents from sustainable tourism.

**Figure 4.** Behaviours adopted by tourists during travel.

In order to analyse the relationship between the respondents' ideas of sustainable tourism and the behaviours they actually adopt, a cross-tabulation analysis was carried out (Figure 5).

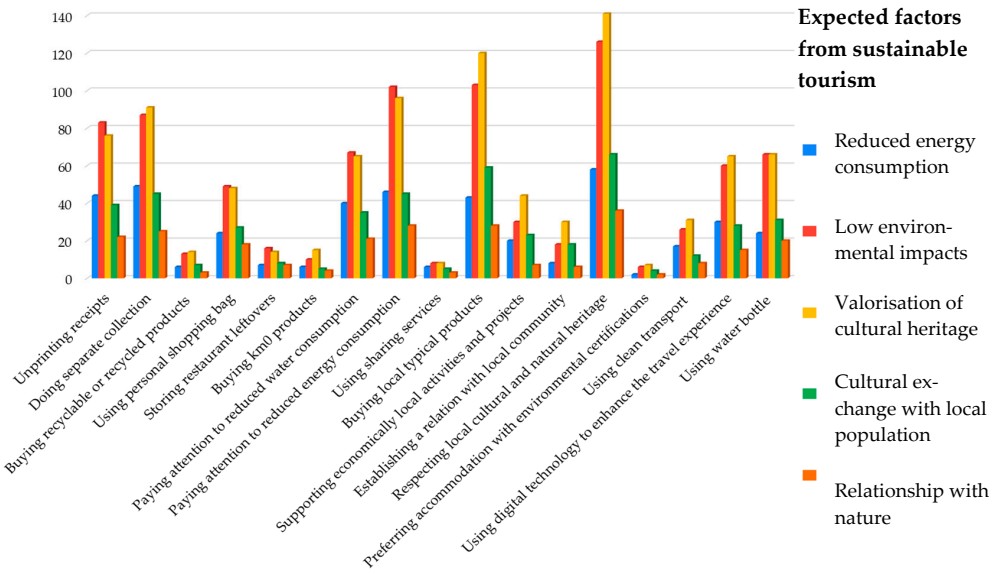

**Sustainable behaviours adopted by respondents**

**Figure 5.** Cross-tabulation between factors that respondents attribute to sustainable tourism (*y*-axis) and sustainable behaviours adopted by themselves (*x*-axis).

This analysis shows that the majority of people who associate sustainable tourism with a reduction in environmental impacts and the valorisation of cultural heritage are also the same who adopt more sustainable behaviour during travel. In fact, among respondents who linked sustainable tourism to a reduction in environmental impacts (141 respondents, representing 65.3% of the total), 89% said that they adopt behaviours that respect the local cultural heritage while travelling, 72% buy typical local products, and 71% are careful about energy consumption (turning off unnecessary lights, using the air conditioning/heating system only when they are in the room, etc.).

Moreover, the issues related to waste reduction are important both for respondents who link sustainable tourism to low environmental impacts and to the enhancement of the local cultural heritage: in fact, for both groups, the most adopted behaviours are separating waste (respectively, 61% for the first group and 55% for the second) and carrying a water bottle during the trip (respectively, 46% for the first group and 40% for the second).

Considering the attention paid to avoiding waste in tourist accommodations, the majority of respondents are "moderately" (42.7%) or "extremely" (30.5%) attentive, while 19.7% are "not at all" and 7% are "slightly" attentive (Figure 6).

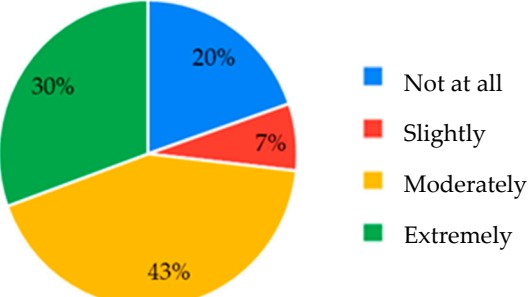

**Figure 6.** Attention paid to avoiding waste at the tourist facilities.

Furthermore, these data (*y*-axis) were cross-referenced with the respondents' age groups (*x*-axis) (Figure 7). The cross-tabulation shows that 25–34-year-olds have a high level of attention to avoiding waste at the facilities where they stay (e.g., in terms of attention to energy saving, water saving, waste management, etc.). However, in this age group,

there is a similarity in results between respondents "extremely" attentive in avoiding waste (24.7%) and those who are "not at all" attentive (27.9%). In contrast, in the other age groups, there is a greater difference between the different levels of attention. In the 55–64 and 65–74 age groups, the percentage of respondents "not at all" attentive to avoiding waste was zero, while 52.9% (in the 55–64 age group) and 64.3% (in the 65–74 age group) of respondents said they were "extremely" attentive to this aspect.

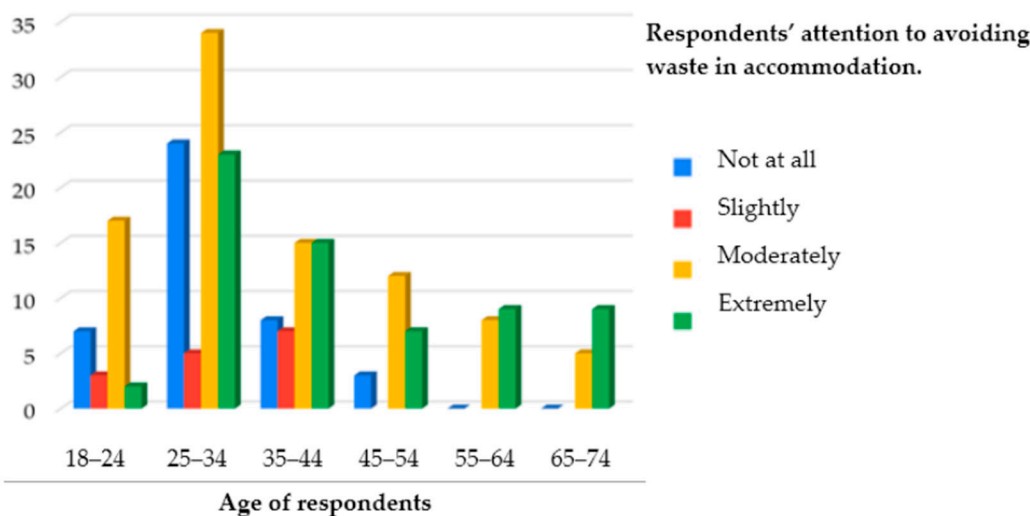

**Figure 7.** Cross-tabulation between respondents' attention to avoiding waste in accommodations (*y*-axis) and their age (*x*-axis).

Concerning the importance attributed to the presence of sustainable services in the choice of the tourist facility (e.g., eco-friendly furniture, organic food, energy produced from renewable sources, etc.), it emerges that this aspect is quite considered, as, for 40% of respondents, this presence is "moderately" important, and for 34%, it is "not at all" important (Figure 8).

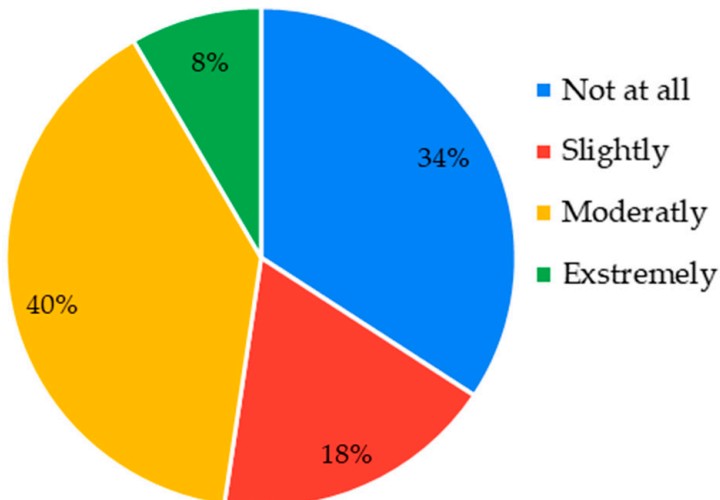

**Figure 8.** Influence of the presence of sustainable services in the choice of the tourist facility.

Considering the means of transport to the travel destination (for this question, more than one answer could be selected) (Figure 9), 153 (out of 216) respondents declare that they prefer walking, while 142 (out of 216) prefer public transport. Only 25 (out of 216) consider the possibility of moving via car-sharing services.

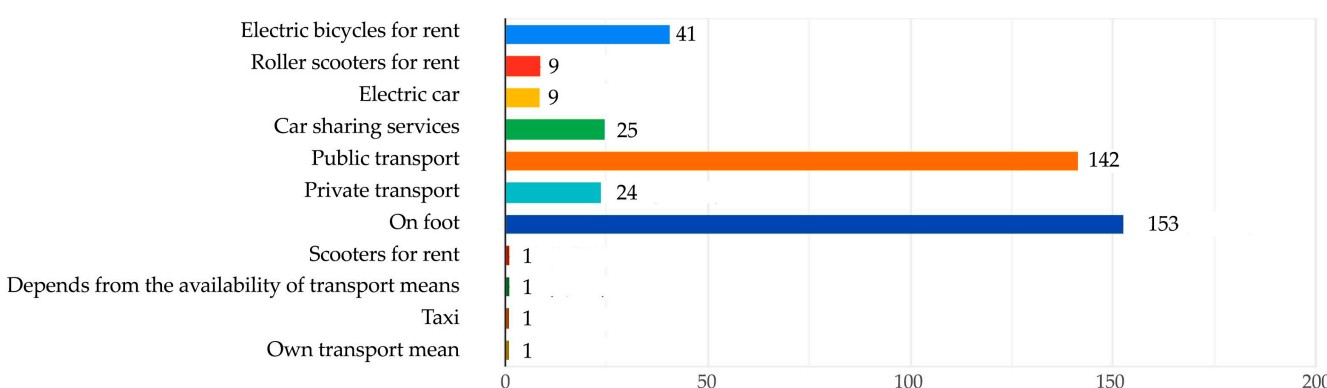

**Figure 9.** Means of transport in the travel destination.

Taking into account the factors influencing the decision to buy products when travelling (Figure 10), there is a predominance of respondents (38%) who prefer goods identified with the destination's brand; 18% prefer goods produced by a manufacturing company with a strong environmental commitment, 15% prefer goods with reduced and/or differentiable packaging, 15% prefer goods produced by a manufacturing company with a strong social commitment, and 14% prefer goods produced with sustainable materials.

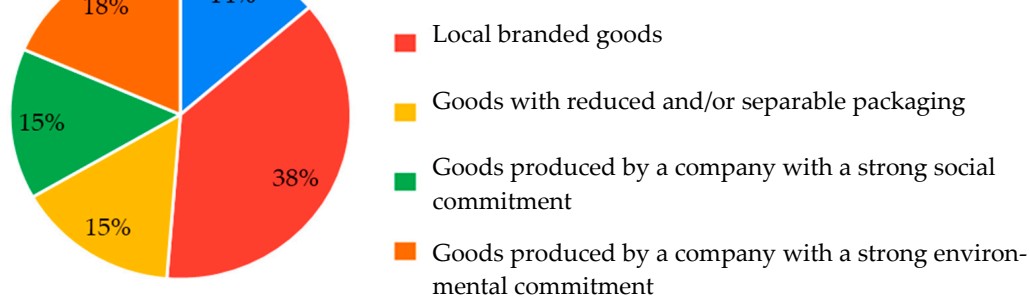

**Figure 10.** Factors influencing the choice to purchase products during travel.

### 4.2.2. Be an "Active and Responsible" Traveller

In the "Be an 'active and responsible' traveller" section, aspects more related to the social dimension of sustainability were addressed. The results are summarised in Table 2 and then analysed in more detail to follow.

**Table 2.** Summary of the results of the third survey section: "Be an 'active and responsible' traveller".

| Be an "Active and Responsible" Traveller | |
| --- | --- |
| Interest of travellers in looking for information about the local culture of the tourist destination | Before the trip, most respondents look for information about the local culture of the tourist destination to be more aware of the place they are going to visit. |
| Interest of respondents about information relating to preventive health measures to be taken before travel | Aspects related to health measures for prevention are particularly felt among the interviewees, as more than one-half of respondents ask about precautionary health measures that should be taken before travel. |
| Relationship between the importance attributed by travellers to a good welcome from the local community and interest in participating in activities with the local community of the tourist destination | A good welcome influences people's readiness to actively participate in local activities. |

**Table 2.** *Cont.*

| Be an "Active and Responsible" Traveller | |
|---|---|
| Relationship between the importance of respondents to be well received by the local community and their interest in participating in activities led by the local community considering respondents who declare they have economically supported local activities or projects | There is a correlation between these three factors. Among those who said they had economically supported local activities/projects, the majority felt it was "extremely" important to have a good welcome from the local community and, at the same time, declared they were "moderately" interested in taking part in activities with the local community. |
| Respondents' interest in participating in activities led by the local community | Slightly less than half of the respondents stated that they were "moderately" interested in participating in activities organised by the local community. |
| Importance attributed by respondents to a good welcome from the local community | Almost the entirety of the respondents considered a good welcome from the local community to be "extremely" important. |
| Frequency of respondents' feedback | The results show that at the end of a trip, just over half of respondents give the feedback "sometimes". |
| Degree of influence of other feedback on respondents' travel choices | Half of the respondents were "slightly" influenced by feedback from other travellers. |
| Relationship between the age of respondents and the frequency of their feedback about tourist experience | The results show that young people (25–34 age group) give feedback much more than other categories, followed by the 35–44 age group. In general, also considering the other age groups, the tourists who have a positive experience leave reviews more than tourists who have a negative experience. |
| Relationship between the frequency of respondents' feedback about their tourist experience and the frequency of the use of social media to share it | Half of the respondents who provided feedback "sometimes" use social media "extremely" to share their experiences. Similarly, among respondents who "always" give feedback, the largest percentage "extremely" share the tourist experience on social media. |

Before the trip, travellers tend to look for information about the local culture of the tourist destination (Figure 11a) both to be more aware of the place they are visiting (84.1%) and to be more respectful of the local lifestyle (12.6%).

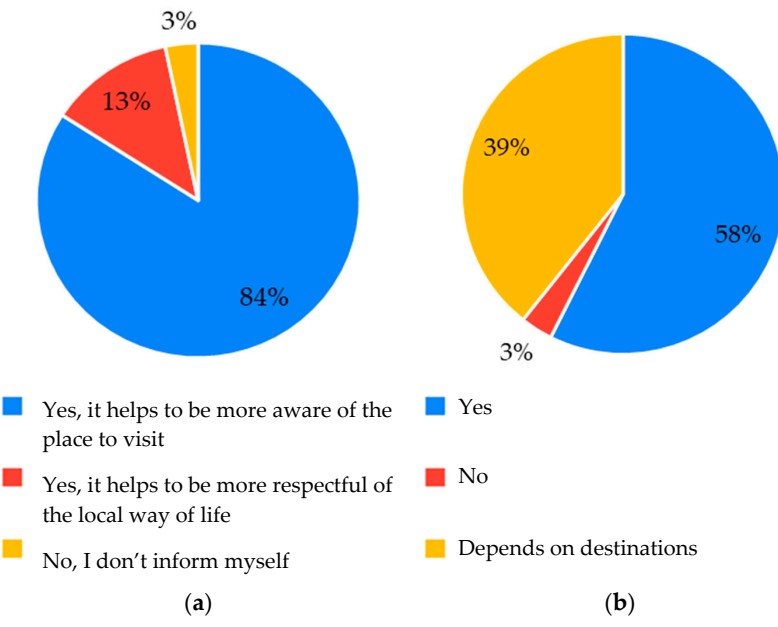

**Figure 11.** (**a**) Interest of travellers in looking for information about the local culture of the tourist destination; (**b**) information sought by respondents on preventive health measures to be taken.

Aspects related to health measures for prevention (vaccinations, healthcare models, etc.)—the result of which is probably influenced by the health emergency we are experiencing—are particularly felt among the interviewees (Figure 11b). In fact, 57.4% of respondents ask about precautionary health measures that should be taken before travel, 39% of respondents declare their decision depends on the destination, and only 3.2% say they have no interest in taking such precautions.

For many travellers, knowledge of the context can also be deepened through dialogue and relationships with locals: in fact, 93.5% of respondents say they stop to talk to locals for this purpose.

From the cross-tabulation analysis relating the importance attributed to a good welcome from the local community (*x*-axis) and the interest in participating in activities with the local community of the tourist destination (*y*-axis), it can be deduced that a good welcome influences people's readiness to actively participate in local activities (Figure 12).

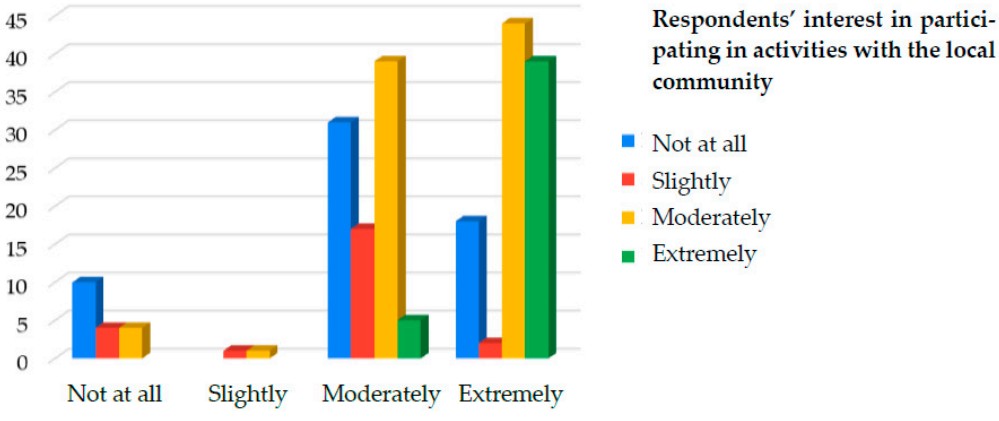

**Figure 12.** Relationship between the importance of respondents being well received by the local community (*x*-axis) and their interest in participating in activities led by the local community (*y*-axis).

A cross-tabulation (Figure 13) was made to understand whether the importance perceived by tourists in receiving a good welcome from the local community and the interest in participating in activities with the local community were related to the travellers' willingness to support local activities or projects economically. In the graph, data are presented only for those respondents who stated that they supported local activities or projects economically. The figure shows that there is a correlation between these three factors as, among those who said they supported local activities or projects economically, 46% said they felt it was "extremely" important to receive a good welcome from the local community, and, among these, 42.7% declared themselves to be "moderately" interested and 37.9% were "extremely" interested in taking part in activities with the local community.

In comparison with other areas of sustainability, the survey's findings indicate that the relationship with the local community of the tourist destination is regarded as being less significant; in fact, only 15.3% of respondents established contact with it (i.e., by taking part in activities organised by the community of the place visited, etc.).

The importance attributed to this interaction is also expressed in "active" terms (as interest in participating in activities organised by the local community) or in "passive" terms (as how the local community welcomes tourists). The results show that the perception of an "active" interaction is considered less important than a "passive" one. In fact, 41.2% of the respondents stated that they were "moderately" interested in participating in activities organised by the local community (Figure 14a), while about 90% of respondents considered a good welcome from the local community to be "extremely" (47.9%) and "moderately" (42.8%) important (Figure 14b).

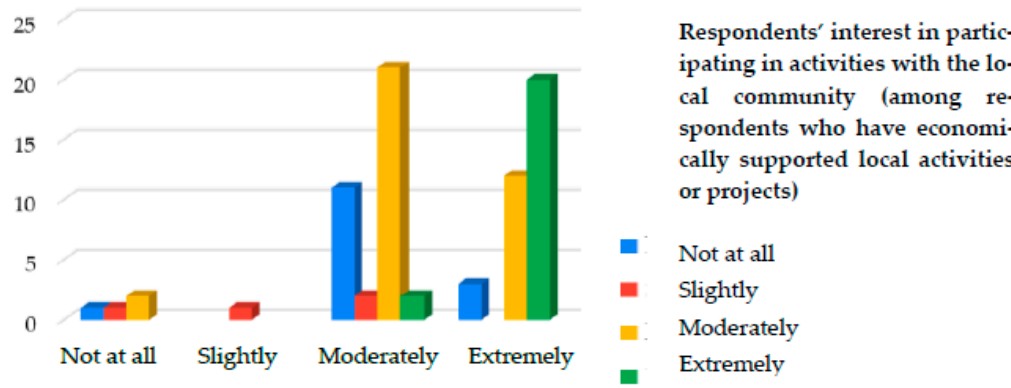

**Figure 13.** Relationship between the importance of respondents being well received by the local community (*x*-axis) and their interest in participating in activities led by the local community (*y*-axis) considering the respondents who declare they have economically supported local activities or projects.

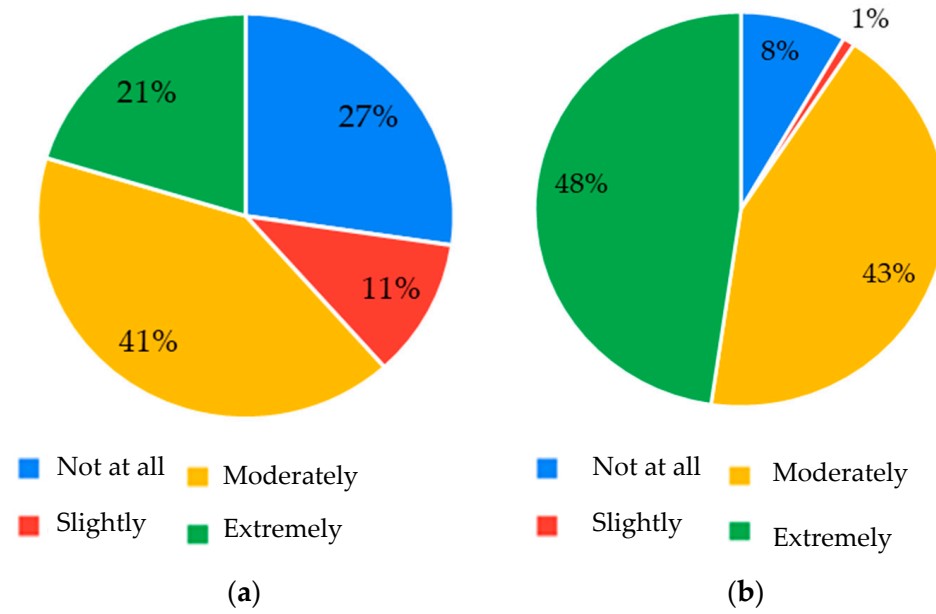

(**a**)    (**b**)

**Figure 14.** (**a**) Respondents' interest in participating in activities led by local community; (**b**) importance attributed by respondents to a good welcome from the local community.

Considering the increasing importance of technology in the enrichment of the tourism experience, the questionnaire investigated the frequency of the use of technology in the respondents' feedback (Figure 15a) and the influence of other travellers' feedback on the respondents' travel choices (Figure 15b).

The results show that at the end of a trip only 21.8% of respondents do not review their trips, while, although not continuously, the remaining percentage give feedback (56.9% leave feedback "sometimes") (Figure 15a). Figure 15b shows that feedback influences the travel choices of many tourists (51.4% are "slightly" influenced, 25.5% are "extremely" influenced, 12% are "moderately" influenced, and 11.1% are "not at all" influenced).

A cross-tabulation analysis was made to understand the frequency of the respondents' feedback at the end of their travel experiences in relation to age (Figure 16). The results show that young people (25–34 age group) give feedback much more than other categories, followed by the 35–44 age group.

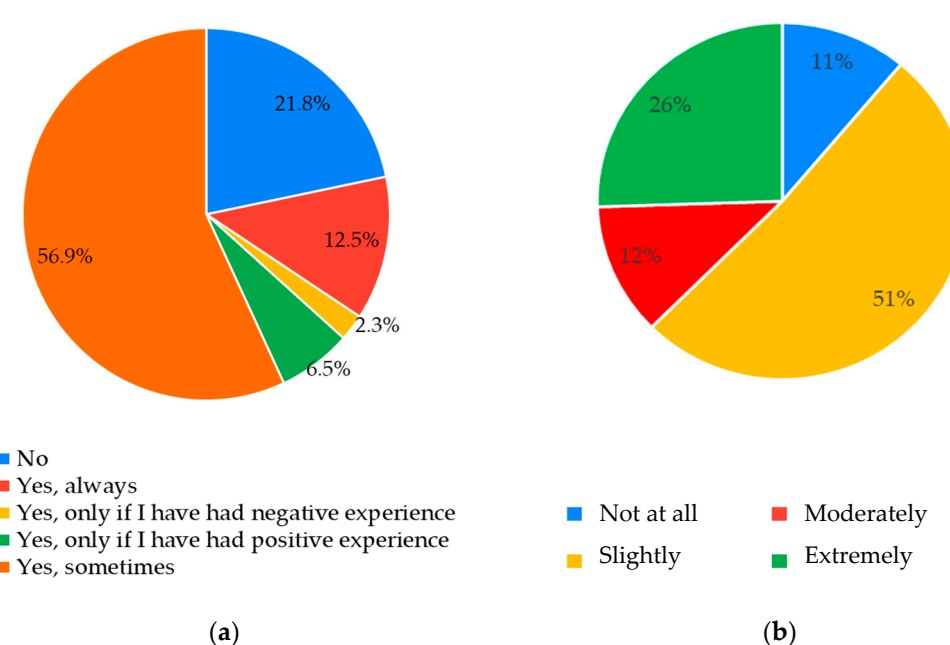

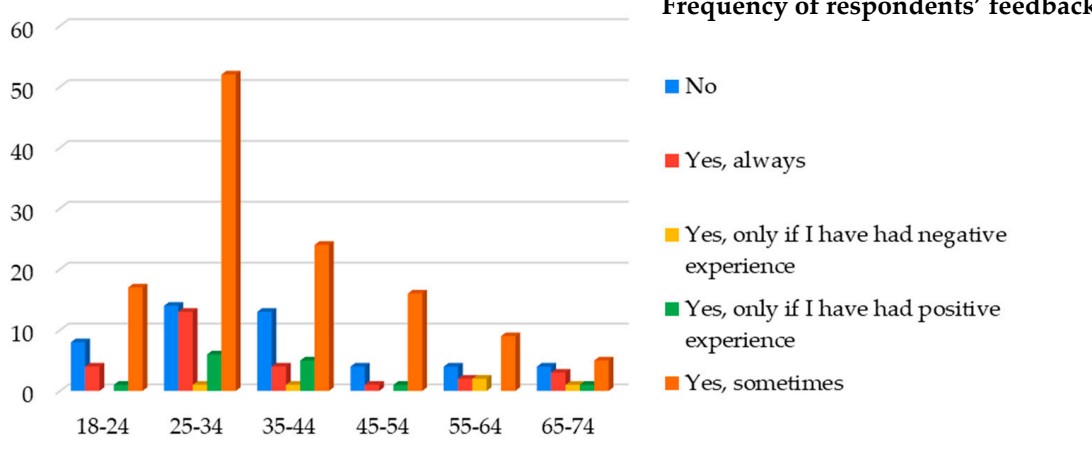

**Figure 15.** (**a**) Frequency of respondents' feedback; (**b**) degree of influence of other feedback on respondents' travel choices.

**Figure 16.** Cross-tabulation between the age of the respondents (*x*-axis) and the frequency of their feedback about tourist experiences (*y*-axis).

In general, also considering the other age groups, tourists who had a positive experience reviewed more than tourists who had a negative experience.

According to a cross-tabulation of the frequency of the respondents' feedback on their travel experiences (Figure 17) and the frequency of the respondents using social media to share their travel experiences (places visited, experiences and activities, products, etc.), among respondents who provided feedback "sometimes" (57%), 29% use social media "extremely" to share their experience. Similarly, among respondents who "always" give feedback (12%), the largest percentage (63%) "extremely" share their tourist experiences on social media.

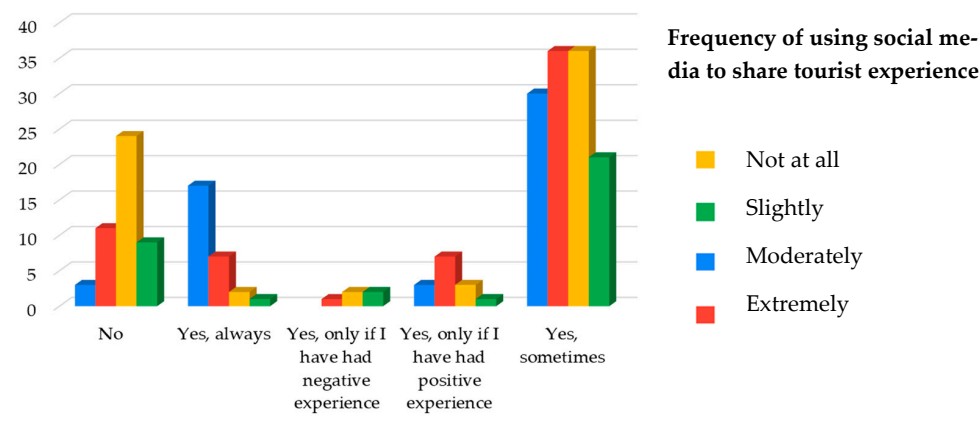

**Respondents' feedback on the tourist experience**

**Figure 17.** Cross-tabulation between the frequency of respondents' feedback about their tourist experiences (*x*-axis) and the frequency of the use of social media for sharing them (*y*-axis).

### 4.2.3. Tourism and COVID-19

The "Tourism and COVID-19" section deals with the influence of COVID on the respondents' sensitivity and choices. The results are summarised in Table 3 and then analysed more in-depth.

**Table 3.** Summary of the results of the fourth survey section: "Tourism and COVID-19".

| Tourism and COVID-19 | |
|---|---|
| Influence of COVID-19 on the choice of tourist destination | The health emergency due to COVID-19 affects the choice of tourist destinations for almost all respondents. Among respondents who declared that COVID-19 influenced their choices, 59.7% would not travel until the health emergency was over. |
| Tourists' perceptions of virtual tourism | The majority of respondents believe that virtual tourism restricts many feelings and perceptions. On the other hand, a small percentage of them believe that virtual tourism can represent a valid alternative to traditional tourism, but only temporarily. |
| Influence of COVID-19 on the respondents' sensitivity to environmental issues | Most respondents are more sensitive to environmental issues after the pandemic, although there is not much difference between percentages of those who are "extremely" influenced and those who are "not at all" influenced. |
| Relationship between influence of COVID-19 on respondents' sensitivity to environmental issues and their age | Only in the 65–74 age group did most respondents state that the pandemic has "extremely" affected their sensitivity to environmental issues. In the other age groups, the "moderate" influence was always predominant, except in the case of the 25–34 age group, in which the percentages of respondents "not at all" and "moderately" affected by the pandemic are almost the same. |
| Relationship between the importance attributed to the "cost of the trip" and the pre-COVID-19 period and post-COVID-19 period | The results show that the importance attributed by the respondents to the cost of the trip significantly changed, as many more people "extremely" paid attention after COVID-19. |
| Relationship between the importance attributed to "health" and the pre-COVID-19 period and post-COVID-19 period | The analysis shows that in the pre-COVID-19 period health security was perceived as a priority travel choice for a small percentage of the respondents. After the first lockdown, however, this percentage tripled, as many of those who had previously answered "slightly" and "moderately" changed their opinion, stating this issue to be "extremely" important. |

The health emergency due to COVID-19 affects tourist activity and the choice of tourist destinations for almost all respondents (Figure 18).

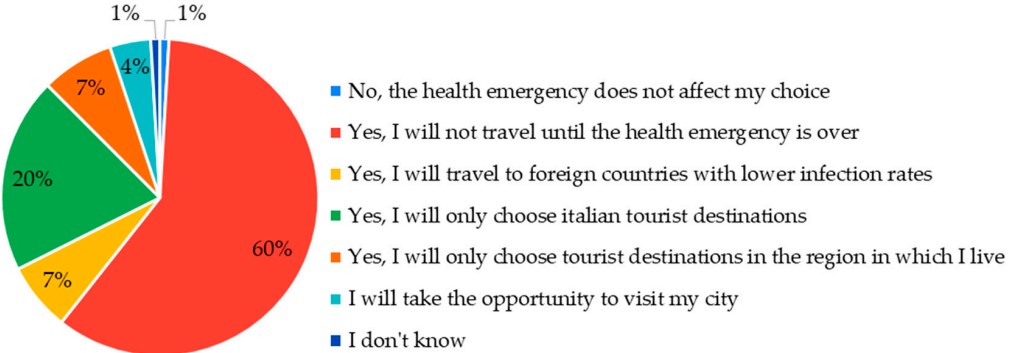

**Figure 18.** Influence of COVID-19 on the choice of a tourist destination.

Among respondents who declared that COVID-19 influenced their choices, 59.7% would not travel until the health emergency was over; 19.9% would continue to travel but only to Italian tourist destinations; 7.4% would continue to travel but only to tourist destinations located in the region where they live; and 6.9% would travel to foreign countries with a lower percentage of contagion. Finally, 4.2% would take advantage of this to become "tourists in their own city".

Since during the pandemic there was an increase in the use of technologies to practice so-called "virtual tourism" [83], the survey investigated tourists' perceptions of the usefulness of this tool. The majority of respondents (66.7%) believe that virtual tourism restricts many feelings and perceptions that, instead, constitute a real-world tourist experience; 38.9% of the interviewees, on the other hand, believe that virtual tourism can represent a valid alternative to traditional tourism, but only temporarily. In total 20.8% of respondents interpret virtual tourism to be a potential factor of exclusion for certain categories of users (the elderly, children, etc.), while only a minority (15.7%) consider virtual tourism to contribute to improving the tourist experience even in "normal" situations (i.e., not in health emergencies).

The pandemic due to COVID-19 was interpreted as anticipating the crisis due to climate change [84], which, if not managed in time, risks becoming disruptive and causing irreversible effects.

Considering the above aspect, a number of questions were formulated in the questionnaire to understand how much the pandemic affected the respondents' sensitivity to environmental issues and, thus, how much the respondents' awareness of the interdependencies between climate change, health, and the pandemic had increased. The results in Figure 19a show that most respondents were influenced, although there is not much difference between the percentages of those who are "extremely" influenced and those who are "not at all" influenced. Furthermore, these data were cross-referenced with the respondents' age groups (Figure 19b). Only in the 65–74 age group did most respondents state that the pandemic has "extremely" affected their sensitivity to environmental issues. In the other age groups, the "moderate" influence was always predominant, except in the case of the 25–34 age group, in which the percentage of respondents "not at all" and "moderately" affected by the pandemic is almost the same.

Finally, the questionnaire investigated how the respondents' perceptions changed in the post-COVID period considering the following criteria for choosing a travel destination: health safety, cost of the trip, the proximity of the destination, the possibility of outdoor excursions, the need for preventive health measures (vaccination, etc.), the possibility of personalising the trip through digital tools (bookings, etc.), the presence of valuable cultural heritage, the presence of valuable natural heritage, and the presence of positive feedback from other tourists. Figure 20a,b show that for the factors "cost of the trip" and "health",

the importance attributed by the respondents significantly changed as many more people "extremely" paid attention after COVID-19.

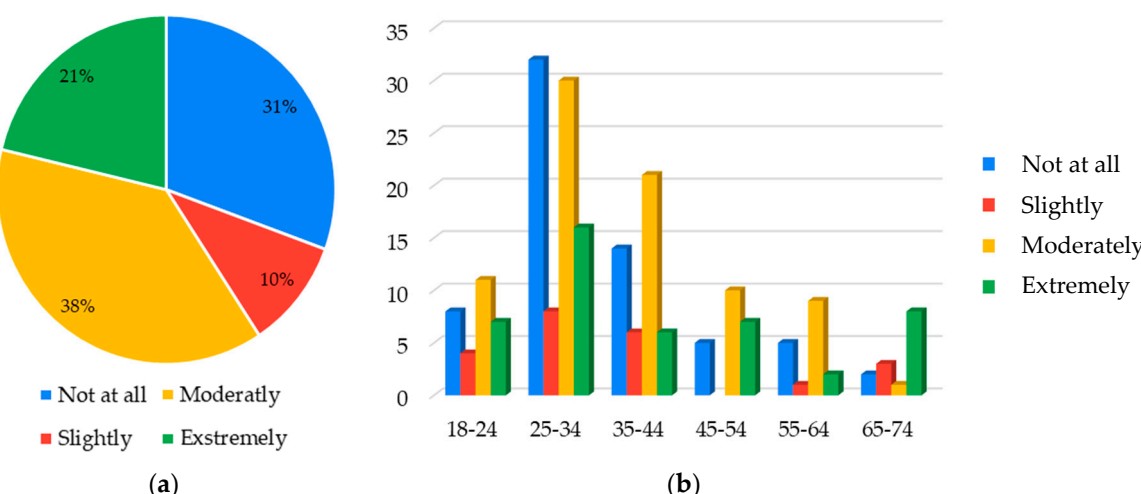

(**a**)         (**b**)

**Figure 19.** (**a**) Influence of COVID-19 on the respondents' sensitivity to environmental issues; (**b**) cross-tabulation between the influence of COVID-19 on respondents' sensitivity to environmental issues and their age.

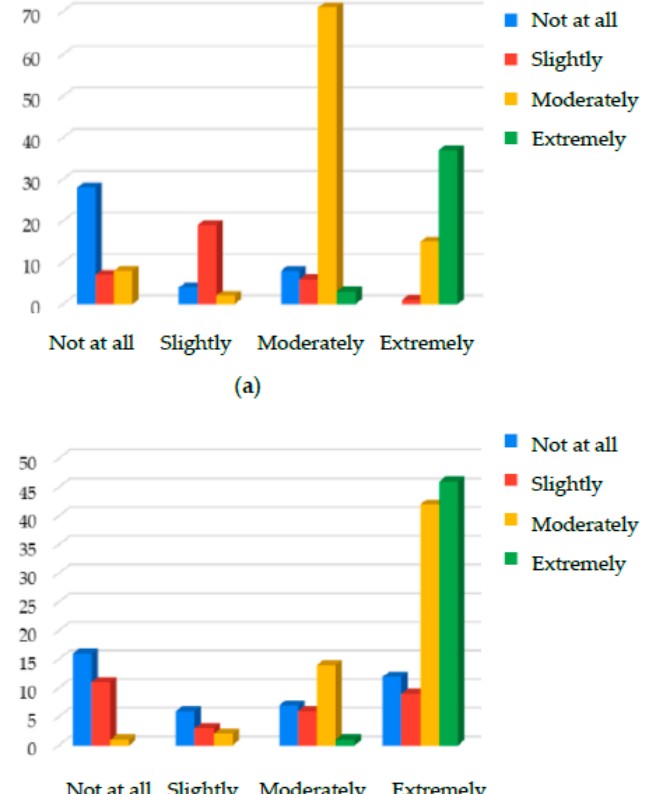

**Figure 20.** (**a**) Cross-tabulation between the importance attributed to the factor "cost of the trip" between the pre-COVID-19 period (*x*-axis) and post-COVID-19 period (*y*-axis); (**b**) cross-tabulation between the importance attributed to the factor "health" between the pre-COVID-19 period (*x*-axis) and post-COVID-19 period (*y*-axis).

## 5. Discussion

The COVID-19 crisis is challenging to different sectors. Among them, the tourism sector has suffered negative impacts, facing a demand shock that is extremely conditioned

by the current health crisis. It has been among the most affected worldwide sectors by the COVID-19 pandemic.

If, on the one hand, it has recorded a negative shock, on the other hand, this crisis has represented (and still represents) a unique opportunity to reflect on important issues about the future of the tourism sector, its resilience, risks, and opportunities. However, in the past, other catastrophic events and crises (the terrorist attack at the World Trade Center in New York on 11 September 2001; the 2003 SARS-CoV epidemic; the world financial crisis of 2008) have shown that tourism is very resilient (tourism websites continue to be used during COVID-19), highlighting how the tourism market has always managed to recover and continue to grow over time.

The COVID-19 situation is changing tourist trends, which also emerged from this survey. The health emergency due to COVID-19 resulted in numerous impacts on the tourism sector, highlighting, even more, the need to operationalise the contribution of tourism to the achievement of the Sustainable Development Goals from the perspective of the circular economy. The analysis conducted in this study starts from the awareness that, in order to orient recovery strategies to overcome the crisis, it is necessary to begin with a change in the behaviour of those involved in the tourism experience, which means a cultural change in considering the relationships between sustainable behaviours and the health of both people and the planet [85,86].

In the last decades of the 20th century, the importance of the involvement of other categories of stakeholders in the decision-making and strategic definition processes of the tourism sector began to emerge [87], especially following the failure of many projects in which the main beneficiaries were excluded from the entire planning phase [88–90].

For some years, some European [91] and non-European countries [85,92] have adopted a policy of openness and dialogue with all stakeholders, activating projects in which the involvement of different stakeholders was the starting point for elaborating shared long-term local tourism strategies and recommendations capable of integrating economic, social, and environmental issues, assuming an open perspective and activating a co-creative process [93,94].

From this perspective, the questionnaire represents a starting point for analysing the behaviour of tourists and the impact of COVID-19 on their perception of sustainability and environmental awareness issues.

The results concerning the respondents' travel motivations and the factors expected by respondents from sustainable tourism show that cultural heritage is a predominant factor both as a motivation and as an element to be valorised from the perspective of sustainable tourism. This highlights that people consider cultural heritage a key factor in sustainable tourism. In fact, European [95,96] and international [9,97] documents on tourism confirm this view and declare that, in order to ensure a long-term perspective for sustainable tourism, it is necessary to focus on increasing travellers' awareness of the value of the cultural heritage of tourism destinations and the importance of linking cultural heritage to sustainable development dynamics [11,98], especially in a post-pandemic situation. In order to positively affect users' awareness, it is necessary to move towards more authentic modes of enjoyment, whose value is manifested in the traveller's immersive participation in activities and experiences made by and with local communities [96], with the support and collaboration of other tour operators, administrations, and bodies in this sector.

Moreover, this relational dynamic between tourists, communities, and places would favour a process of collective capacity-building [99] capable of strengthening the sense of attachment and common interest in the preservation and care of this heritage, stimulating a realignment of behaviour with respect to dynamics that are more respectful of places and people and, for this reason, more sustainable. The recent study by [100] statistically demonstrated that the greater the identity value expressed by cultural heritage, the more sustainable the consumption pattern of travellers is. This trend was confirmed by the survey results, showing that there is a correlation between the three factors related to the importance perceived by tourists in receiving a good welcome from the local community, the

interest in participating in activities with the local community, and the tourist's willingness to support local activities or projects economically. For many travellers, the knowledge of a tourist destination can also be deepened through dialogue and relationships with local people, as almost all respondents stated that they stop to talk to local people to this end.

The analysis of the results suggests an important reflection on the importance of educating the local community to welcome the tourist, a factor that is able to ensure more authentic experiences and stimulate the respect and valorisation of local culture through a more authentic and immersive relationship with local people. This would have two advantages: on the one hand, it would foster a greater understanding of the "spirit of place" [101–103] from tourists' point of view and, on the other hand, it would stimulate their "sense of belonging" to the place [104,105], stimulating them to know more about local activities and projects. Indeed, research has shown that the more tourists interact with local communities, the more awareness of their values and trust in the projects and activities they conduct increases [106–109]. As demonstrated by the ONMEST 2 project (See "OMNEST 2—Open Network for Mediterranean Sustainable Tourism 2" project https://erfc.gr/projects/onmest2/; accessed on 13 July 2022), developing "full immersion" tours first implies a training phase for local stakeholders involved in the experience in order to develop sustainable tourism activities and communicate a lasting message to users beyond the experience. The same suggestion regards aspects more closely related to tour operators, which will also be key to the recovery of the sector. Knowledge exchange (e.g., related to good hospitality practices) and clear and transparent information (generating trust) will certainly be key issues. In addition, training courses will be important so that tour operators' proposals can be aligned with new emerging market demands.

The above considerations confirm the role of communities as a way to achieve sustainable tourism [109,110], contributing both to the enforcement of local identity as a "brand" [111] and to more sustainable planning and development in tourism destinations [112]. In relation to this aspect, the survey evaluated the attitude of tourists towards supporting local production [89]. The data confirm the trend recorded in recent studies [90–92] that attributes greater attractiveness to products with local brands than those produced by large retail chains. In fact, as reported in the study, "It tastes better because . . . consumer understandings of UK farmers' market food" [93]; consumers are attracted to small-scale local markets because the craftsmanship of the production process guarantees a higher product quality and, at the same time, contributes to the enhancement of the local cultural heritage based on traditional knowledge and skills.

The product's sustainability in terms of the producer's actions and the impact it has on the environment during the stages of manufacture, packaging, and delivery is another crucial consideration. This aspect is, on the one hand, connected to the previous factor (as many people believe that by choosing locally produced food they reduce the environmental impact due to transport [90,94–98]) and, on the other hand, it is linked to consumer sensitivity in favouring companies that adopt both environmental and social sustainability policies, ensuring that, in addition to reducing their carbon footprint, they also guarantee safe working conditions, contributing to new employment opportunities and supporting disadvantaged groups [99].

As COVID-19 has profoundly undermined this aspect of the direct relationship between tourists and local people, preventing the physical contact that is the basis of all the most authentic and immersive experiences, buying local products or, more generally, using technological platforms, have become the most frequently used ways of supporting productive communities and small- and medium-sized enterprises in difficulty, developing a sense of proximity that, although virtual, is capable of manifesting itself in concrete effects. This is why, during the pandemic period, there was an increase in the use of technology to convert processes that traditionally take place on-site and de visu with digitised processes. Today, technology is playing an increasingly important role in the management of tourism supply and demand. Especially in recent times, technology has been adopted in

tourism practices to stimulate interaction and mutual exchange between different tourism stakeholders, contributing to the enrichment of the tourism experience [113,114].

This is an important aspect in the perspective of sustainability as it not only contributes to generating and regenerating relationships [11,71] but also increases knowledge about the tourist experience (i.e., exchanging feedback with other users about the travel experience, searching for in-depth information about places to visit, etc.). Indeed, technology can be used for different but interconnected purposes: on the one hand, it can be implemented to support tourist services and, on the other hand, it facilitates the creation of new tourist experiences [115], favouring the sharing of knowledge and information [116,117] and also of emotions and experiential moments [118]. In some experiences (https://www.feelflorence.it/it; accessed on 18 June 2022; https://www.iamsterdam.com/it; accessed on 18 June 2022; https://www.holidaytravelreports.com/Travel/Banyumas.aspx; accessed on 18 June 2022), technology helps visitors experience unusual itineraries, supporting them to better organise their stays. Furthermore, the use of technology in tourism practices stimulates interaction and mutual exchange between different tourism stakeholders, thus enriching the tourism experience [119–121] and improving the management of tourist assets to increase their attractiveness (see the Heland Project (http://geredis-society.org/heland/; accessed on 2 August 2022). This relational aspect "brings" the different actors closer together, stressing the idea of the tourist as a prosumer [122,123] and changemaker [124,125].

The increasing potentiality of technologies that allow more and more digital interaction, also in an anonymous way [126], encourages the tourist's contribution to building a sense of identity [127] and strengthening cultural capital through knowledge or ideas. Furthermore, this aspect favours the development of virtual communities, creating new forms of social interactions and ties [128].

In social terms, the above relationships refer to the collaborative and cooperative relationships among tourists and between tourists and host communities. In cultural terms, they refer to the knowledge exchange between the different tourist actors. Finally, in economic terms, they refer to the contribution and support of tourists to the local economy [105].

More generally, the data about the use of social media to share tourist experiences show that, in various ways, this behaviour is becoming increasingly common [129–131], especially as a means of knowledge [132]. The widespread use of platforms (such as TripAdvisor, Booking, etc.) and social features that allow people to share their experiences in "live" mode is determined by their ability to provide authentic information free from advertising requirements or other commercial interests. This advantage increasingly makes sharing a knowledge tool that is used by users both "actively" (as content producers) and "passively" (as users).

The COVID-19 pandemic has highlighted the fragility of the current development model and made clear the need to initiate a transition towards sustainability through the implementation of circular processes.

New investments and a reconsideration of priorities in the context of recovery from COVID-19 present unique opportunities for shaping healthier environments and scaling up actions accordingly. The prescriptions of the WHO Manifesto highlight the importance of an integrated recovery strategy in which the tourism sector assumes a key role in ensuring economic and social prosperity together with the enhancement of well-being and respect for the environment.

In a short time, there has been a marked reduction in pollution, leading to a rapid improvement in the quality of the environment (more clean air and waters), and technology has been used as a tool to speed up the processes of working and connecting with each other, reshaping our lifestyles in a more flexible way and reducing the environmental impacts of our activities (i.e., pollution and congestion due to travelling to work). Still, today, there is a growing awareness among people that they need to ensure that natural resources are protected to ensure that the ecosystem does not collapse again and that strategies are truly effective in the medium-to-long term.

Furthermore, the current situation has led to the emergence of new needs and new trends related to the behaviours and habits of tourists, such as safety, hygiene, social distancing, etc.

While international tourism is recovering more slowly, domestic tourism continues to lead the industry's recovery. According to experts, the importance attributed by tourists to aspects of safety and hygiene over economic convenience is directing tourism towards new dynamics that are defined as "domestic tourism" [133,134], which is one of the main travel trends that will continue to shape tourism in 2022 [135]. As the results of the questionnaire also showed, COVID-19 has influenced and will continue to influence tourists in choosing, for example, a tourist destination in relation to the distance from their place of residence. Thus, proximity is an important aspect of tourism recovery (at least in an early stage of recovery). People prefer to book stays in nearby places or will even prefer to become "tourists in their own town".

These trends are allowing for the revaluation of the relationships of proximity with both places and people based on slower rhythms of use and more authentic experiences. Furthermore, recent studies [24,135,136]—stimulating the growing demand for open-air and nature-based tourism activities, with increasing interest in domestic tourism and "slow travel" experiences—are much more focused on the valorisation and promotion of the peripheral areas (Handbook to Tourism Projects—Hungary-Croatia IPA CBC Programme 2007–2013) (see NEWPER Project (http://www.enpicbcmed.eu/sites/default/files/newper_4.pdf; accessed on 9 July 2022), FOP—Future of Our Past project (https://www.uni-med.net/progetti/fop/; accessed on 12 July 2022).

This trend is confirmed by the results of the questionnaire and is interesting because it may represent a new sector on which to focus recovery even though, although it is driving the recovery of several destinations, in most cases, it is only partially offsetting the drop in international demand [4]. For this reason, until today, many tourists consider virtual tourism to be just a temporary solution that cannot replace the emotions and perceptions offered by a tourist experience in loco. Furthermore, virtual tourism represents a potential factor in the exclusion of certain categories of users (the elderly, children, etc.), denying the inclusive principle of sustainable tourism.

Once the pandemic crisis is over, people are likely to be much more willing to engage in outdoor activities and will also prefer tourism experiences that provide a certain level of safety in their enjoyment (aspects of sanitation, cleanliness, social distancing, and general safety during their tourism experiences), reducing participation in overcrowded activities and destinations.

Considering this aspect, the survey results confirm the growing tendency of so-called bio-contributive travel [137], which is represented by a set of behaviours (such as buying typical local products, being careful about energy consumption, etc.) that express travellers' desire not only to do no harm to the environment but to personally and actively (and consciously) contribute to reducing negative impacts (for example, making carbon-positive choices) [138,139]. This trend is confirmed by the offers of destinations typically recognised as "Ecotourism Destinations" (i.e., Malaysia, the Galapagos Islands, Ecuador, Finland, Morocco, Greece, etc.) in which the tourist has the opportunity to experiment with ecotourism itineraries to get closer to nature without damaging the environment. These kinds of activities, besides conveying a meaningful ecological and cultural message, produce concrete benefits not only for the environment but also for the whole economic and social system of the tourist destination. Indeed, the active participation of tourists in eco-activities managed by local people contributes both to the economic development and dynamism of local communities. At the same time, in the last few decades, more and more projects have been dealing with ecotourism (The International Ecotourism Society https://ecotourism.org/project-summaries/; accessed on 9 July 2022; Cambodia Sustainable Landscape and Ecotourism Project https://projects.worldbank.org/en/projects-operations/project-detail/P165344; accessed on 11 June 2022), demonstrating that by enhancing the awareness and knowledge on ecotourism through increased con-

nections between all stakeholders involved in the sector it has the potential to create new jobs and learning opportunities, thus really contributing to the sustainable development of territories.

The lack of knowledge about ecotourism and, more generally, the attitude towards considering waste management, energy, and water consumption to be the main environmental aspects of sustainable tourism, is confirmed by the small percentage of respondents who said they chose accommodations with environmental certifications. For this reason, more and more studies [112] and projects (https://www.shmile2.eu/shmile2-en.html; accessed on 28 July 2022) are spreading around the world to demonstrate the benefits of the eco-label, stressing their potential to become a marketing opportunity in terms of attractivity for more conscious tourists and also in terms of the development of a new, highly qualified employment sector.

In truth, it emerges that this aspect takes on greater weight for respondents when considered in absolute terms as a potential factor influencing the choice of a tourist facility than when compared to other aspects that can be translated into concrete attitudes. This trend is probably also linked to the excessively complex and technical aspects that characterise such certifications, thus making it difficult to understand the benefits of such certifications for non-"technical" users.

Another aspect about which tourists are not very aware is the choice of means of transportation. COVID-19 certainly influences tourists' preferences in choosing means of transportation that ensure social distancing or otherwise reduce the chances of contagion. From this perspective, tourists prefer (for the moment) to travel by private means to reach tourist destinations. Equally, to get around within the tourist destination, rental transportation systems (such as bicycles and scooters) that combine the speed of transportation and compliance with anti-COVID-19 safety standards will be preferred.

This aspect is influenced by the geographical (and, consequently, cultural) context of the tourists, as well as their lack of awareness of the environmental effects of this factor. In fact, it is evident that Italy, compared with other countries, is currently slower in the transition towards sustainable forms of mobility [140] and sharing services. Furthermore, spacing and sanitation provisions have certainly influenced the preference for private transport rather than sharing services.

The lack of awareness among tourists about the environmental impacts of their means of transportation is also a consequence of a lack of knowledge and dissemination about existing sustainable practices: A greater awareness of these issues would also help to make these attitudes more widespread and give a considerable boost to the transition already underway. Finally, a factor that certainly inhibits and slows down the choice of sustainable modes of transport, especially in the case of electric mobility, is the high cost of these services. Although environmental awareness is growing among tourists, it is undeniable that it is only feasible if the economic compromise is satisfactory. The sustainability of tourist experiences also translates into economic sustainability, and imposing too high a cost for sustainable services necessarily implies the exclusion of certain categories of users, once again making the awareness-raising process dependent on economic rather than cultural dynamics.

## 6. Conclusions

As stated in the previous sections, the crisis caused by COVID-19 has also demonstrated that dividing the three dimensions (ecological, economic, and social) has been a huge mistake. This has forced us to rethink the current economy, linking it more to the economies of ecology and society. Starting from this consideration, COVID-19 also makes us rethink the tourism enterprise and business models, highlighting the financial and economic impacts, but also the social and environmental impacts. The tourist enterprise has to be useful not only to the local economy but also to society and nature. The pandemic has been interpreted as anticipating the climate change crisis [84], which, if not managed in time, risks becoming disruptive and causing irreversible effects. The systemic structure of

these two crises highlights the multidimensional relationships and interactions existing both between them and between social, natural, and economic systems [84].

The COVID-19 pandemic highlights the need to "correct" the ecological paradigm from a more humanistic perspective [141], recognizing the common value between these two paradigms: the "intrinsic values" of the natural ecosystem and of the human being [102,142–146]. In this way, the ecological paradigm promotes the full capabilities of the human being [147] and guarantees the centrality of human rights.

On this basis, human beings and their well-being have to be placed at the centre of the development strategies of the tourism sector in a medium–long-term perspective, guaranteeing the rights and needs of future generations according to the new human and ecological paradigm.

When the circular economy model is implemented in the tourism sector, the processes of wealth creation are intertwined with the import capacity (attractiveness to tourists, visitors, talents, and capital) and the export capacity (handicrafts, art, local identity products, and knowledge products) [148].

From this point of view, the circular tourism model provides a new theory of values that includes both instrumental value and other types of value, such as intrinsic value. In addition, the COVID-19 situation leads us towards this new concept of value.

The links and synergies between the various actors that are directly or indirectly involved and work together to achieve a shared goal constitute the "heart" of the circular tourism concept rather than a single actor. Therefore, each actor and stakeholder has the power to affect the decisions and processes (and thus have responsibility) [11].

This relational dimension, which characterises the circular model, introduces the concept of co-responsibility, in which the responsibility of each is linked with that of others. From this perspective, the promotion of consumers' co-responsibility has a central role in the enhancement of the awareness of consumers about the purchase and use of more sustainable products and services [149,150]. Co-responsibility requires considering all stakeholders (individual companies, private institutions, communities, etc.) in the evaluation process.

Furthermore, the awareness of tourists (but also of all stakeholders and actors of tourism) is fundamental to making the transition to Human Circular Tourism operational, and thus, awareness programs for tourists should be elaborated [151].

The first step towards the effective implementation of circular tourism is to clarify the concept itself, identifying a common language not only to better elaborate development strategies but also to facilitate communication. This common language refers both to the concepts themselves and to tools for operationalizing them (i.e., criteria and indicators of evaluation tools). An awareness of the benefits that such a model can produce is crucial for its effective implementation. The behaviour of the different stakeholders and actors involved depends the success (or otherwise) of the HTC model. For this reason, from a research perspective, some guidelines to orient the behaviours of the different stakeholders involved are necessary.

Furthermore, in order to contribute to better awareness and knowledge about Human Circular Tourism, knowledge-sharing plays a central role. It is also important to identify and share good circular tourism practices to understand the factors of success (or failure) and replicate them (or not) in other experiences. This is important not only to allow operators to inspire and implement (where possible) similar strategies, but it is also important to increase consumers' awareness and direct their choices and behaviour towards a more sustainable perspective.

From a more general perspective, as already implemented in other countries [43,85,150,152, 153], guidelines for the implementation of Human Circular Tourism should be included in official policies. Tourism could be more integrated into national and local planning in order to orient the strategic development of cities in the long-term, identifying the synergistic relationships and intersections between the tourism sectors and other ones, considering

that the tourism sector is deeply interconnected with and dependent on several resource flows and value chains (i.e., construction, finance, retail, and agriculture).

The current health emergency forces us to find new solutions in the short term (mainly related to the tourist supply), as well as in the medium and long term. This brings attention not only to the consumers but also to the producers, as well as to other actors in the tourism sector. This is the reason why the methodology presented in this paper needs to be replicable in other tourist actor categories (companies, academia, donors, international organisations, and public bodies).

The results of this study are intended to be a useful support for decisionmakers in guiding choices about strategies and actions to move towards more sustainable and circular tourism. The success (or failure) of such strategies and actions also depends on the behaviour of tourists, and, therefore, their views are important in understanding how to orient them.

However, a limitation of this study is represented by the interviewed sample, which, of course, represents a small proportion of stakeholders. The interviewed sample includes people from all over the world. However, it should be noted that the respondents' answers may be influenced by external factors, such as lifestyle, social condition, the political situation in the countries where they live, etc.

A future research step may be aimed at expanding the category of stakeholders surveyed, extending the survey (in addition to additional tourists) to other stakeholders involved in the identification and implementation of strategies and actions for more circular and sustainable tourism (UNWTO categories). In addition, a system of indicators to evaluate these strategies is necessary to understand the extent to which they actually contribute to producing benefits from this perspective.

**Author Contributions:** This paper is a result of joint work. However, it is possible to attribute as follows: Conceptualisation, M.B. and F.N.; methodology, M.B. and F.N.; background and Human Circular Tourism concept, F.N.; survey development, M.B. and F.N.; data curation, M.B.; writing—original draft preparation, M.B. and F.N.; writing—review and editing, M.B. and F.N.; critical analysis of the results, M.B. and F.N.; funding acquisition, F.N. All authors have read and agreed to the published version of the manuscript.

**Funding:** This research was founded by the Ministry of Education, University and Research (MIUR) under the Research Project of Relevant National Interest (PRIN) program (Project title: "Metropolitan cities: territorial economic strategies, financial constraints and circular regeneration") grant number [2015STFWFJ_004].

**Institutional Review Board Statement:** Not applicable.

**Informed Consent Statement:** We submitted the questionnaire to more than 200 people and stated at the beginning of the questionnaire that the data would be treated anonymously and for scientific and research purposes.

**Data Availability Statement:** The data are those of the results we reported in the paper.

**Conflicts of Interest:** The authors declare no conflict of interest. The funders had no role in the design of the study; in the collection, analyses, or interpretation of data; in the writing of the manuscript; or in the decision to publish the results.

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
