# Peer review of "Human Circular Tourism as the Tourism of Tomorrow: The Role of Travellers in Achieving a More Sustainable and Circular Tourism"

_sustainability, doi:10.3390/su141912218_

Round 1
Reviewer 1 Report
Dear Authors,
In the paper entitled “The tourism of tomorrow: the Human Circular Tourism at the COVID-19 era“ is proposed as strategy to move towards a more sustainable future and thus to reduce the negative impacts produced by the tourism sector.
The presented research results and recommendations may contribute to the elaboration and implementation of strategies and actions towards a more sustainable and circular tourism.
After reading the paper, I have comments and suggestions to improve the paper as follows:
Abstract
I propose to correct the abstract according to the Journal "Sustainability". There is no information about the methods used and the results of the study are not presented.
Titel
Title of the work should be modified. It lacks reference to the area of research. The content of the article does not fully correspond to the title of the paper.
Structure of Article
I suggest improving the structure of the article according to the guidelines of the journal.
New chapter numbering should be introduced
1. Introduction
2. Literature review/Theoretical background
3. Materials and Methods
4. Results
5. Disscusion
In Introduction
The introduction to the topic is interesting and based on world literature.
At the end of this chapter, the purpose of the paper should be stated. The paper does not ask detailed research questions.
[101-109] - this information is redundant, does not contribute anything
Literature review/theoretical background
The following chapters are too much [110-435]. Please select the content in terms of the topic of the paper.
[191]- in this line appears information about the purpose of the article
“In this paper, the circular tourism model is proposed as a model for making tourism more sustainable, capable of producing environmental, social and economic benefits at the same time”
In Materials and Methods
[445]- the next purpose of the work is given:
“In this study, a survey has been conducted involving community through interviews aiming at understanding their awareness about sustainable and circular tourism, their behaviours as tourists and the influence of the COVID-19 on their traveller’s choices.”
In order to better understand the purpose of the work, I suggest to introduce a scheme of research procedure.
The Results
The results were presented and described in a good way, they are very interesting and important for the development of sustainable tourism. However, I would suggest doing some organizing of the obtained results in the form of tables. The text needs to be clear and easy to read.
In the Discussion section, the authors should discuss and explain the conclusions and results of the work more. The authors should compare their project and results with the results of similar ongoing research on this topic from other parts of Europe and around the world.
This section should still answer the question: what tangible benefits has this study brought to the development of sustainable tourism?
All in all, I recommend this paper for publication in the Journal “Sustainability” after major changes.
Kind regards,
Reviewer
Author Response
Dear reviewer,
we have attached below the file with the responses to your comments.
Thank you, kind regards

Reviewer 2 Report
This has potential to be a very useful paper which raises, and proposes solutions for, the need for a well coordinated transnational strategic approach to tourism development. However, it is very much a Eurocentric view and if global issues such as climate change are to be integrated there would seem to be a need for a more global strategy. Certainly many of the themes (including those that have been enshrined in buzzwords like ‘glocalisation’ in other literatures) are canvassed here, but what really jumped out was the extent to which the focus is urban. The attempt to integrate economic, social, cultural etc themes is very welcome.
Rural and wilderness tourism is, for many parts of the world, a major part of the tourism sector and one that requires different strategic approaches because of the very different resourcing needs. Equally, many of the cited climate-endangering problems cited are predominantly rural based (agriculture and forestry are obvious examples) but intrinsically linked to urban consumption. While this paper does recognise the ‘cultural and natural heritage’ dimensions in tourism, the understanding of these seem a little shallow.
There are some really good points in section 6 that could well have been expanded upon in more depth. This would have improved the balance of the paper and strengthened section 7. The paper also has some very useful references. As it stands, it feels out of balance. Perhaps if the first sections were slightly more coherent and efficiently handled and the discussion strengthens the paper would be much stronger. There is a hint of an underdeveloped theme that would seem to be central to the paper (hinted at by the 'vector versus victim" thought)
The use of pie charts is interesting. Fig 1 also is intriguing … what is the difference between a ‘worker’ and an ‘employee’, for example? Also for a topic where motivations are notoriously multidimensional the way these results are presented is rather too simplistic. Fig 3 is one where a pie chart is simply the wrong type of graph to use because people will have multiple behaviours and this gives no sense of the range and complexities of the behaviours. Moreover a snapshot of behaviours tells us little unless we also have some sense of whether these are changing over time and, if so, in what way. Simply dichotomising it as pre- and post- covid does not really work when the intention is to develop a strategy that sits across multiple complex shocks.
Motivations matter – e.g avoiding waste might be as a strategy for sustainability, it might be a traditional habit or it may well be to maximise available resources. What this reviewer is struggling with is to understand how sustainability awareness impinges on actions because the citied factors (local cultural environment, health preventative measures etc) are all things that have driven travellers/tourists for decades and meaningful engagement with local communities has always been, at best, superficial (tourists tend to be consumers rather than producers).
A little repetitive in places so perhaps an edit for efficiency might prove useful, especially if it helped to clarify and sort some of the more muddled ideas. The English is generally fine but an edit for grammar, syntax and readability would be advised. This is very important in those sections where it is not exactly clear what the authors are advocating.
I would encourage the authors to attend to the basics for this paper and perhaps pick up on some of the points above to pursue in some follow up studies. I am vacillating between a 'minor' and a 'major' revision for this paper - as it stands it is rather too muddled for easy reading but does cover a topic of some importance - I would strongly recommend a thorough edit. Generally the English is not too bad, so that is not the problem. The problem seems to be clarity of the ideas.
Author Response

(The authors gave the same response as above.)

Round 2
Reviewer 1 Report
The article has been revised in accordance with the reviewer's comments. I do not make any comments.
Reviewer 2 Report
The authors are to be thanked for the careful consideration of reviewer comments. This is now a much better paper on a very important topic. The amendments have all been carefully considered and added appropriately. My only additional suggestion (which I meant to make in the first instance) is that it might be useful (if feasible) to append the questionnaire used because this will be helpful to future researchers as well as readers of this paper. I am mindful that many times I have caught myself asking "but what was actually asked?" when reading older research papers.